# Orbital Transformers for Predicting Wavefunctions in Time-Dependent Density Functional Theory

**Xuan Zhang**[1]    **Haiyang Yu**[1]    **Chengdong Wang**[2]    **Jacob Helwig**[1]
**Shuiwang Ji**[1,2,3*]    **Xiaofeng Qian**[2,4,5*]
[1]Department of Computer Science and Engineering, Texas A&M University
[2]Department of Materials Science and Engineering, Texas A&M University
[3]J. Mike Walker '66 Department of Mechanical Engineering, Texas A&M University
[4]Department of Electrical and Computer Engineering, Texas A&M University
[5]Department of Physics and Astronomy, Texas A&M University
{xuan.zhang, sji, feng}@tamu.edu

## Abstract

We aim to learn wavefunctions simulated by time-dependent density functional theory (TDDFT), which can be efficiently represented as linear combination coefficients of atomic orbitals. In real-time TDDFT, the electronic wavefunctions of a molecule evolve over time in response to an external excitation, enabling first-principles predictions of physical properties such as optical absorption, electron dynamics, and high-order response. However, conventional real-time TDDFT relies on time-consuming propagation of all occupied states with fine time steps. In this work, we propose OrbEvo, which is based on an equivariant graph transformer architecture and learns to evolve the full electronic wavefunction coefficients across time steps. First, to account for external field, we design an equivariant conditioning to encode both strength and direction of external electric field and break the symmetry from SO(3) to SO(2). Furthermore, we design two OrbEvo models, OrbEvo-WF and OrbEvo-DM, using wavefunction pooling and density matrix as interaction method, respectively. Motivated by the central role of the density functional in TDDFT, OrbEvo-DM encodes the density matrix aggregated from all occupied electronic states into feature vectors via tensor contraction, providing a more intuitive approach to learn the time evolution operator. We adopt a training strategy specifically tailored to limit the error accumulation of time-dependent wavefunctions over autoregressive rollout. To evaluate our approach, we generate TDDFT datasets consisting of 5,000 different molecules in the QM9 dataset and 1,500 molecular configurations of the malonaldehyde molecule in the MD17 dataset. Results show that our OrbEvo model accurately captures quantum dynamics of excited states under external field, including time-dependent wavefunctions, time-dependent dipole moment, and optical absorption spectra characterized by dipole oscillator strength. It also shows strong generalization capability on the diverse molecules in the QM9 dataset. Our dataset is available at https://huggingface.co/divelab, and our code is available as part of the AIRS library https://github.com/divelab/AIRS/.

## 1 Introduction

Density functional theory (DFT) (Hohenberg & Kohn, 1964; Kohn & Sham, 1965) provides an efficient way to solve time-independent many-body Schrödinger equation using a variational principle and has been widely applied to compute the properties of the ground state of molecules and solids. However, many important physical and chemical phenomena involve the excited states and the dynamic responses of the systems to external perturbations. In such cases, time-dependent

---

*Equal senior authorship.

density functional theory (TDDFT) (Runge & Gross, 1984) provides a natural extension of the time-dependent many-body Schrödinger equation. It can be formulated and solved in frequency space in linear-response TDDFT (Casida, 1995), or in the time domain via real-time TDDFT (RT-TDDFT) (Runge & Gross, 1984; Yabana & Bertsch, 1999; Qian et al., 2006; Ullrich, 2011), enabling the investigation of excited state properties such as excitation spectra, optical absorption, charge transfer, and electron dynamics under time-dependent external fields such as electromagnetic fields. Starting from the static electronic wavefunctions obtained within ground-state DFT, RT-TDDFT propagates these wavefunctions in the time domain under the influence of an external field, allowing direct investigation of both linear and nonlinear physical properties.

However, RT-TDDFT is computationally demanding due to the temporal and spatial discretization of Kohn-Sham wavefunctions, long-time propagation, repeated evaluations of the Kohn-Sham Hamiltonian, and the increasing number of Kohn–Sham wavefunctions with system size. To accelerate this procedure, machine learning (ML) provides a promising way to replace or approximate the costly propagation steps, thereby accelerating quantum dynamical simulations while retaining accuracy (Zhang et al., 2025). In this work, we propose a new model, OrbEvo, designed to learn the full wavefunction evolution while incorporating the underlying physical symmetries of the TDDFT problem. In particular, we consider the SO(2) equivariance induced by the presence of an external field, and we demonstrate how ML-based partial differential equation (ML-PDE) frameworks can be adapted to capture quantum dynamics effectively. We extend PDE learning to the setting of wavefunction coefficient evolution on atom graphs, while enforcing SO(2) equivariance to respect the system's symmetry constraints. Furthermore, we propose effective methods to handle multiple electronic states, which remain agnostic to the choice of backbone neural architecture. Together, these innovations allow our approach to bridge the gap between *ab initio* quantum dynamics and scalable ML-based approximations.

## 2 PRELIMINARIES

In this section, we will provide a formulation of the RT-TDDFT problem. At the same time, the constraints inherent to this physical problem will be elaborated on, serving as the motivation for the techniques developed. Our method is built upon and enabled by existing literature. We review them in Appendix A.

**DFT with predefined localized atomic orbital basis set.** DFT provides a practical approximation to solve the many-body Schrödinger equation of a molecular or material system. Instead of explicitly modeling the many-body wavefunctions, DFT represents the system using a set of single-particle Kohn-Sham wavefunctions $\{\psi_n \colon \mathbb{R}^3 \to \mathbb{C}\}_{n=1,\dots,N_{\text{occ}}}$, where $N_{\text{occ}}$ denotes the number of occupied electronic states. Each electronic state can be occupied by up to two electrons according to the Pauli exclusion principle. To construct these Kohn-Sham states, DFT often employs a basis set, such as the localized atomic orbitals in this work, $\{\phi_o \colon \mathbb{R}^3 \to \mathbb{C}\}_{o=1,\dots,N_{\text{orb}}}$, with $N_{\text{orb}}$ the total number of orbitals in the system. These atomic orbitals are spatially localized around atoms and describe the electronic states of isolated atoms, forming the Hilbert space of the system. In the linear combination of atomic orbitals (LCAO) method, each electronic wavefunction $\psi_n$ can be expressed as a linear combination of atomic orbitals, $\psi_n = \sum_{o=1}^{N_{\text{orb}}} \mathbf{C}_{no} \phi_o$, where $\mathbf{C} \in \mathbb{C}^{N_{\text{occ}} \times N_{\text{orb}}}$ is the coefficient matrix defining the contribution of each orbital. At the ground state, the coefficients are determined by solving the Kohn-Sham equation (Kohn & Sham, 1965) in the matrix form, denoted as

$$\boldsymbol{H}\mathbf{C}_n = \epsilon_n \boldsymbol{S}\mathbf{C}_n, \tag{1}$$

where $\boldsymbol{H} \in \mathbb{C}^{N_{\text{orb}} \times N_{\text{orb}}}$ is the Kohn-Sham Hamiltonian matrix, $\boldsymbol{S} \in \mathbb{R}^{N_{\text{orb}} \times N_{\text{orb}}}$ is the overlap matrix, and $\epsilon_n \in \mathbb{R}$ are the eigen energies for the Kohn-Sham eigen states. This formulation highlights the central role of the Hamiltonian and overlap matrices in determining the electronic structure.

**TDDFT under external electric field.** For the TDDFT problem in this paper, the input consists of atom types and 3D atomic positions of the molecule, denoted as $\mathbf{z} \in \mathbb{N}^{N_a}$ and $\mathbf{R} \in \mathbb{R}^{N_a \times 3}$, respectively, where $N_a$ is the number of atoms in the system, together with an applied time-dependent uniform external electronic field $\mathbf{E}(t) \in \mathbb{R}^3$, as well as the initial ground state wavefunction coefficients $\mathbf{C}(0)$. The goal is to predict the temporal evolution of the electronic wavefunction, represented by a sequence of coefficient matrices $\{\mathbf{C}(t)\}_{t=1}^T$ that reconstruct the wavefunctions at each time step.

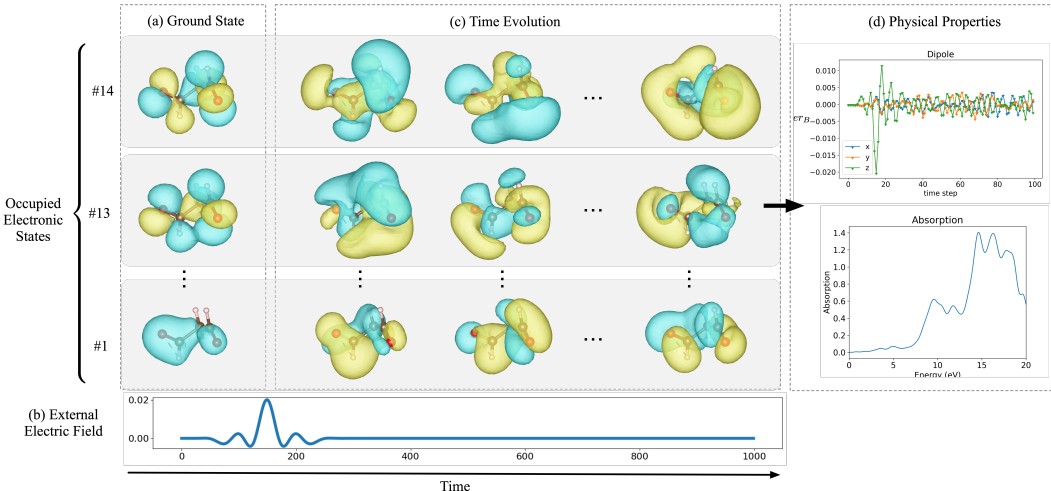

Figure 1: The framework of RT-TDDFT. (a) Ground state wavefunctions as the initial input. (b) External electric field applied onto the system. (c) Time evolution of wavefunctions under external field. (d) Physical properties calculated from the time-dependent wavefunctions and dipole moments.

In the absence of external electronic field, the dynamics reduces to simple unitary evolution over time, $\mathbf{C}_n(t) = \exp(-i\epsilon_n t/\hbar)\mathbf{C}_n(0)$, $n = 1, \ldots, N_{\text{occ}}$, corresponding to phase rotations of the electronic wavefunction. However, under a time-dependent electronic field $\mathbf{E}(t)$, the perturbation couples on these electronic wavefunctions, leading to nontrivial transitions that must be captured by the time-dependent Kohn–Sham equations in the LCAO basis as follows,

$$\frac{d}{dt}\mathbf{C}_n(t) = -\frac{i}{\hbar}\boldsymbol{S}^{-1}\boldsymbol{H}(t)\mathbf{C}_n(t), \tag{2}$$

where $\boldsymbol{H}_{oo'}(t) = \langle\phi_o(t)|\hat{\boldsymbol{H}}(t)|\phi_{o'}(t)\rangle$, $\hbar$ is the Planck constant, and $\hat{\boldsymbol{H}}(t)$ is the Kohn-Sham Hamiltonian operator at time $t$, given by $\hat{\boldsymbol{H}}(t) = \hat{\boldsymbol{T}}_{\text{el}} + \hat{\boldsymbol{H}}_{\text{H}}[\rho(\mathbf{r}, t)] + \hat{\boldsymbol{V}}_{\text{XC}}[\rho(\mathbf{r}, t)] + \hat{\boldsymbol{V}}_{\text{ext}}(t)$. Within LCAO, the time-dependent electron density is $\rho(\mathbf{r}, t) = \sum_{o=1}^{N_{\text{orb}}} \sum_{o'=1}^{N_{\text{orb}}} \boldsymbol{D}_{oo'}(t)\phi_o(\mathbf{r})\phi_{o'}^*(\mathbf{r})$, where $\boldsymbol{D}$ is the density matrix given by $\boldsymbol{D}_{oo'}(t) = \sum_{n=1}^{N_{\text{occ}}} \eta_n \mathbf{C}_{no}(t)\mathbf{C}_{no'}^*(t)$. $\eta_n$ is the occupation number in electronic state $\psi_n$. The central task of RT-TDDFT is therefore to integrate Equation 2 over time, compute wavefunction coefficients $\mathbf{C}(t)$ in the local orbital basis, subsequently calculate electron density $\rho(\mathbf{r}, t)$, update the density-dependent operators in the Kohn-Sham Hamiltonian and compute $\boldsymbol{H}_{oo'}(t)$, and repeat this process iteratively for many time steps. In RT-TDDFT, each Kohn-Sham wavefunction $\psi_n$ evolves in time under the time-ordered evolution operator $\hat{U}(t, t_0)$, starting from the initial time $t_0$: $\psi_n(t) = \hat{U}(t, t_0)\psi_n(t_0)$, where $\hat{U}(t, t_0) = \hat{\mathcal{T}}\exp\left(-\frac{i}{\hbar}\boldsymbol{S}^{-1}\int_{t_0}^t \hat{\boldsymbol{H}}(t')dt'\right)$, and $\hat{\mathcal{T}}$ is time-ordering operator. More details about time evolution of wavefunctions can be found in Appendix G. For the machine learning model, the objective is to learn the time evolution of the Kohn–Sham wavefunctions $\psi_n(t)$, or equivalently $\mathbf{C}_n(t)$, in order to accelerate TDDFT calculations.

**SO(2) equivariance in TDDFT.** While property prediction, force field prediction, and Hamiltonian matrix prediction are typically formulated under SO(3) equivariance, meaning that when the input geometry is rotated, the corresponding predicted properties transform consistently under the same rotation, this full rotational symmetry can be broken in the presence of an external field. In particular, when a uniform external electronic field along a specific direction is introduced, it defines a preferred spatial direction. As a result, rotations that modify the angle between the field direction and the molecular orientation will alter the system, whereas rotations around the field axis preserve SO(2) equivariance. Consequently, the overall symmetry of the system is reduced. In this work, we focus on the case of a uniform external electronic field applied along a specific direction, where the molecular system is SO(2) under rotations around its axis, thereby reducing the symmetry requirement for predicted properties from SO(3) to SO(2) equivariance to consider the effect of uniform external electronic field. The SO(2)-equivariance of TDDFT data is tested in Appendix H.2, Figure 7.

## 3 METHOD

### 3.1 OVERALL FRAMEWORK

The overall problem framework for TDDFT is illustrated in Figure 1. We describe the inputs and targets of this framework, along with the multi time step outputs strategy used during both training and inference of our machine learning model.

**Delta transformation for capturing small changes in wavefunction coefficients.** One particular challenge in our data is how to define the prediction target. Due to the small magnitude of external electric field, the coefficients at future time steps differ only by a small amount compared to the initial step by the factor of a global phase. Directly learning the wavefunction coefficient will make the model only learn the global phase changes. To correctly model the delta wavefunction, we define a global phase factor and delta coefficients for each electronic state as

$$\gamma_n(t) = \frac{\mathbf{C}_n(0)^\dagger \boldsymbol{S} \mathbf{C}_n(t)}{|\mathbf{C}_n(0)^\dagger \boldsymbol{S} \mathbf{C}_n(t)|} \in \mathbb{C}, \quad \Delta_n(t) = \frac{1}{\beta}\left(\frac{\mathbf{C}_n(t)}{\gamma_n(t)} - \mathbf{C}_n(0)\right) \in \mathbb{C}^{N_{\text{orb}}}, \quad (3)$$

with $\beta = 1 \times 10^{-3}$ to amplify the delta, in which case we have $\mathbf{C}_n(t) = (\mathbf{C}_n(0) + \beta\Delta_n(t))\,\gamma_n(t)$. Note that $\mathbf{C}_n(0)$ is real-valued and the conjugation takes no effect. In the absence of external electric field, i.e., when $\mathbf{C}_n(t) = \exp(-i\epsilon_n t/\hbar)\mathbf{C}_n(0)$, we will obtain $\gamma_n(t) = \exp(-i\epsilon_n t/\hbar)$ since $\mathbf{C}_n(0)$ is real-valued, and $\Delta_n(t) = \mathbf{0}$. This highlights that the proposed delta transformation is able to extract the delta wavefunctions induced by the external electric field $\mathbf{E}(t)$. Since the $\Delta(t)$ carries the most information related to physical properties, we focus on learning $\Delta(t)$ in the main text, while the learning of the $\gamma(t)$ can be found in Appendix K.

**Time bundling.** Time bundling (Brandstetter et al., 2022) is a technique in PDE surrogate models. Instead of advancing time by one at each prediction step, we predict multiple future time steps at once so that the total number of auto-regressive steps will be reduced to produce the same number of total time steps. Formally, our model learns the mapping

$$\mathcal{M}(\theta) : \mathbf{C}(0), \Delta(t-h), \dots, \Delta(t-1) \;\mapsto\; \Delta(t), \dots, \Delta(t+f-1), \quad (4)$$

where $\mathcal{M}$ is the neural network with parameters $\theta$, $h$ is the number of conditioning steps, and $f$ is the number of future steps. We use $h = f = N_{\text{tb}} = 8$ in our implementation.

By using neural networks to approximate the time propagation process, the simulation time can be greatly reduced compared to classical numerical solvers. For example, the simulation time for one molecule using TDDFT solver would take hours, compared to $\sim$1 second for neural network inference. Given the predicted wavefunction coefficients, we can then calculate the properties of the molecule, including dipole moments and absorption spectra.

### 3.2 MODEL

#### 3.2.1 EQUIVARIANT GRAPH TRANSFORMER

Our model is based on EquiformerV2 (Liao et al., 2024), which is an SO(3)-equivariant graph transformer, and we use SO(2)-equivariant electric field conditioning to break the symmetry to SO(2). In EquiformerV2, each node of the graph has an equivariant feature $\mathbf{f}_i \in \mathbb{R}^{d_{\text{sph}} \times d_{\text{emb}}}$ where $d_{\text{sph}}$ is the number of spherical channels and $d_{\text{emb}}$ is the embedding dimension. The spherical channels are partitioned into different segments where each segment has a different rotation order $\ell \geq 0$. The rotation order $\ell$ defines the equivariance property of each segment when the global reference frame of input space undergoes a 3D rotation, and an order-$\ell$ segment has $2\ell + 1$ spherical channels, indexed by $m \in [-\ell, \ell]$. For example, when the input reference frame is rotated by a rotation matrix $\mathcal{R} \in \mathbb{R}^{3\times3}$, then $\ell = 0$ features will transform as scalars and remain unchanged, $\ell = 1$ features will transform as 3D vectors and will be rotated by the same matrix $\mathcal{R}$, and $\ell = 2$ features will transform as order-2 spherical harmonics and will be rotated by the corresponding wigner-D matrix $\mathfrak{D}(\mathcal{R}) \in \mathbb{R}^{5\times5}$. Although EquiformerV2 has the possibility of reducing the range of $m$ to be smaller than $[-\ell, \ell]$, we always use the full $2\ell + 1$ spherical channels in our implementation.

Equivariant graph transformers are composed of equivariant transformer blocks, which process the features with equivariant graph attention and node-wise feedforward networks. The key operation

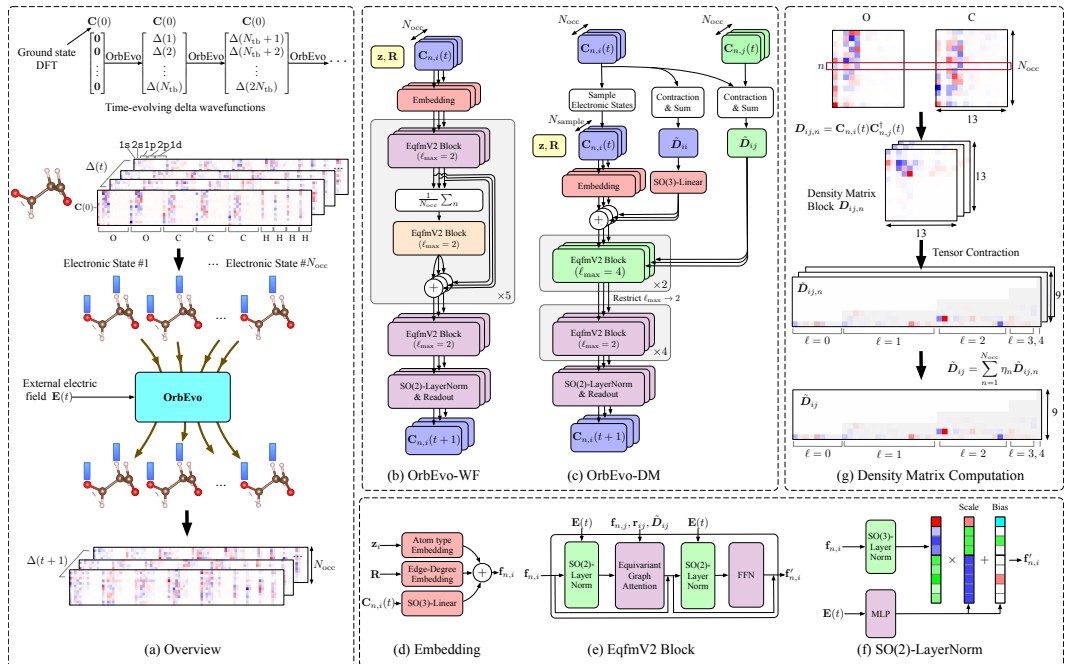

Figure 2: **(a)** Overview of OrbEvo. **Top:** Given the molecular structure and ground-state wavefunctions, OrbEvo predicts the delta wavefunctions (Equation 3) in future steps (one time bundle) autoregressively. **Bottom:** OrbEvo takes wavefunction coefficients as node features on 3D atom graphs, where each electronic state is represented by one graph. The output node features correspond to the target wavefunction coefficients at the next time bundle. **(b, c)** OrbEvo architectures. **(b)** OrbEvo-WF uses layer-wise pooling and global transformer blocks to perform electronic state interactions. **(c)** OrbEvo-DM computes density matrix features from input wavefunctions via tensor contraction and linear projection. Diagonal block features are added into node features and off-diagonal block features are conditioned in equivariant graph attentions. **(d)** Embedding layer, where atom type embedding, edge degree embedding and linear projection of input coefficients are added together. **(e)** EquiformerV2 block with SO(2) equivariance, composed of two SO(2)-LayerNorm layers, one equivariant graph attention layer and one feed forward network. **(f)** SO(2)-LayerNorm, where the output of the SO(3)-LayerNorm in the original EquiformerV2 is multiplied by a *scale* vector and added with a *bias* vector. The *scale* and *bias* vectors are computed from the external electric field intensity at current and the next time bundles with an MLP. *Scale* has different values for different rotation order $\ell$'s, which preserves the SO(3) equivariance. *Bias* has non-zero values only at $m = 0$, which breaks the symmetry from SO(3) to SO(2). **(g)** Illustration of density matrix featurization via tensor contraction.

in equivariant graph attention is to compute a rotation invariant attention score $\alpha_{ij}$ and a rotation equivariant message $\mathbf{m}_{ij}$ between node $i$ and its neighbor node $j$. $\alpha_{ij}$ and $\mathbf{m}_{ij}$ are computed using tensor products between the concatenated node features $[\mathbf{f}_i, \mathbf{f}_j]$ and the spherical harmonics projection of their relative vector $\mathbf{r}_{ij}$ as $\alpha_{ij}, \mathbf{m}_{ij} = \text{TP}_\theta([\mathbf{f}_i, \mathbf{f}_j], \mathbf{r}_{ij})$, where $\text{TP}_\theta$ contains parameters that encode the distance information and mix different rotation orders. Node $i$'s feature is then updated as the weighted sum of messages $\mathbf{f}'_i = \sum_{j \in \mathcal{N}(i)} \alpha_{ij} \mathbf{m}_j$, where $\mathcal{N}(i)$ denotes the node $i$'s neighbors.

### 3.2.2 WAVEFUNCTION GRAPHS WITH SHARED GEOMETRY

We model the wavefunctions on atom graphs where each atom has as feature its atom type $z_i \in \mathbb{N}$ and its coordinates $\mathbf{r}_i \in \mathbb{R}^3$.

**Wavefunction as node features.** The wavefunction coefficients for atomic orbitals of the same atom are grouped together to form initial wavefunction features. The coefficients are further grouped according their rotation orders $\ell$. The resulting wavefunction feature for electronic state $n$ and atom $i$ is $\mathbf{f}_{n,i}^{\text{WF}} \in \mathbb{R}^{d_{\ell 2} \times d_{\text{cond}}}$, where $d_{\ell 2} = 9$ corresponds to the concatenation of rotation orders up to $\ell = 2$, and the $d_{\text{cond}} = 2(2N_{\text{tb}} + 1)$ corresponds to the concatenation of real and imaginary parts of the

conditioning $N_{tb}$ steps and the initial state $\mathbf{C}(0)$, which is real-valued. The additional multiplicative factor of 2 is the multiplicity of rotation orders, which accounts for the fact that each atom has two $\mathtt{s}$ orbitals and up to two $\mathtt{p}$ orbitals. Since each atom has zero or one $\mathtt{d}$ orbital in our data, we use zero padding to fill the second multiplicity channel of rotation order $\ell = 2$. We also zero-pad atoms with fewer orbitals to the same maximum rotation order and multiplicity, which practically only affects hydrogen atoms, which have orbitals $\mathtt{1s}$, $\mathtt{2s}$ and $\mathtt{1p}$.

**Electronic states as set of graphs.** As the wavefunctions of all occupied electronic states jointly decide the electron density, and consequently the propagation operator, it is important to consider the interaction between electronic states when evolving each individual electronic state. One straightforward option would be ordering the electronic states according to their energy levels $\{\epsilon_n\}_{n=1,...,N_{occ}}$ and concatenating all electronic states together into a global feature vector. However, as shown in Appendix B, we find such an approach fails to learn the propagation. We attribute this failure to the fact that the electronic states are eigen vectors of the initial Hamiltonian matrix and are better interpreted as a set, thus mixing them as separate feature channels would make learning difficult.

Instead, we propose to model each electronic state as individual graphs $\{\mathcal{G}_n\}_{n=1,...,N_{occ}}$ where $\mathcal{G}_n = \{\mathbf{F}_n^{WF}, \mathbf{z}, \mathbf{R}\}$. $\mathbf{F}_n^{WF} = \{\mathbf{f}_{n,i}^{WF}\}_{i=1,...,N_a}$ is the node features of electronic state $n$. $\mathbf{z}$ and $\mathbf{R}$ are atom types and coordinates shared by all electronic states.

**Wavefunction encoding.** We apply a linear layer to $\mathbf{f}_{n,i}^{WF}$ and increase its number of channels from $d_{cond}$ to $d_{emb}$, where different weights are used for different rotation order $\ell$'s, and bias is added to $\ell = 0$. We also add the atom type embedding and the edge degree embedding from EquiformerV2 (Liao et al., 2024) to the projected wavefunction features.

### 3.2.3 LEARNING INTERACTION OVER ELECTRONIC STATES

We introduce two ways to interact among electronic states.

**Interaction via wavefunction pooling.** Following set learning methods (Qi et al., 2017; Maron et al., 2020), we do average pooling after each graph transformer block over electronic states. The pooled feature is processed with another graph transformer block and is subsequently broadcasted back to each individual electronic states. Formally,

$$\mathbf{f}_i^{pool} = \text{GT}\left(\frac{1}{N_{occ}}\sum_{n=1}^{N_{occ}}\mathbf{f}_{n,i}\right),\tag{5}$$

$$\mathbf{f}_{n,i}' = \mathbf{f}_{n,i} + \mathbf{f}_i^{pool},\tag{6}$$

where GT is a graph transformer block (as in EquiformerV2).

**Interaction via density matrix.** We use tensor product contraction to extract features from diagonal and off-diagonal blocks of the density matrix. The density matrix is defined as $\boldsymbol{D}(t) = \sum_{n=1}^{N_{occ}} \eta_n \mathbf{C}_n(t) \otimes \mathbf{C}_n^*(t) \in \mathbb{C}^{N_{orb} \times N_{orb}}$, where $\otimes$ is the outer product between vectors. We divide the density matrix into matrix blocks $\boldsymbol{D}_{ij}$ according to which atom pairs the left and right coefficients in the outer product belong to. We then use tensor contraction to re-organize each $\boldsymbol{D}_{ij}$ matrix into a set of equivariant features with rotation orders up to $\ell = 4$. In practice, we use the linearity of tensor contraction to implement this process by first computing the atom pair features for each electronic state as

$$\tilde{\boldsymbol{D}}_{ij,n} = \text{TC}\left(\mathbf{C}_{n,i}(t) \otimes \mathbf{C}_{n,j}^*(t)\right).\tag{7}$$

The tensor contraction operation TC flattens the matrices into equivariant feature vectors via a change of basis using Clebsch-Gordan coefficients. We then aggregate over electronic states and compute the density matrix feature as

$$\tilde{\boldsymbol{D}}_{ii} = \sum_{n=1}^{N_{occ}} \eta_n \tilde{\boldsymbol{D}}_{ii,n}, \quad \tilde{\boldsymbol{D}}_{ij} = \sum_{n=1}^{N_{occ}} \eta_n \tilde{\boldsymbol{D}}_{ij,n}.\tag{8}$$

The resulting high-order features $\tilde{\boldsymbol{D}}_{ii}$ and $\tilde{\boldsymbol{D}}_{ij}$ describe the density matrix blocks for the self-interaction of each atom and the interactions between pairs of different atoms, respectively. An illustration for the density matrix feature computation is shown in Figure 2(g). Additional information on tensor product contraction can be found in Appendix H.1. Due to delta transform, the

density matrix will contain both linear term and quadratic term on delta wavefunctions. We find that including the quadratic term will hurt the performance (as shown in Appendix L), potentially due to its small contribution in the density matrix which may be more sensitive to noise, we thus only keep the linear term in our model. The diagonal pairs of the density matrix are linearly projected and added to the initial node features and the off-diagonal density matrix features are projected into the same channels using linear layers and are used in computing the graph attention, denoted as $\alpha_{ij}, \mathbf{m}_{ij} = \mathrm{TP}_\theta([\mathbf{f}_i, \mathbf{f}_j, \mathrm{linear}(\tilde{\boldsymbol{D}}_{ij})], \mathbf{r}_{ij})$.

### 3.2.4 ORBEVO MODELS

We design two OrbEvo models based on the above two interaction methods. The model architectures are shown in Figure 2.

**OrbEvo-WF.** The model uses pooling as electronic state interaction. It has 6 local graph transformer blocks, each followed by pooling and a global graph transformer block except for the last layer, resulting in 5 global blocks in total. The model is called full wavefunction model as it makes use of wavefunction features from all electronic states at each layer.

**OrbEvo-DM.** The model uses density matrix interaction. The density matrix is computed from the input coefficients. The model has 6 layers in total, the off-diagonal blocks are feed into the first two layers of the model. We use $\ell = 4$ for the first two layers and $\ell = 2$ for the later 4 layers since the computational cost associated with higher-order features is much higher. The feature conversion from $\ell = 4$ to $\ell = 2$ is done by only keeping the lower $\ell$ features.

**Electronic state sampling.** Since OrbEvo process different electronic states in parallel, and the number of electronic states grows linearly with the number of atoms, the computational cost of OrbEvo will be the cost of processing one molecular graph using the backbone equivariant graph transformer multiplied by the number of electronic states. This can increase the training cost significantly, particularly for larger systems. To mitigate this, we do sampling on the electronic states during training and only supervise on the sampled electronic states. As a result, only a subset of electronic states will be processed by the network layers during training. We indicate the electronic state sampling using suffix -s. For example, WF-sall means we use all electronic states when training OrbEvo-WF, and DM-s8 means we randomly sample 8 electronic states when training OrbEvo-DM. We find that sampling will degrade the performance of the full wavefunction model significantly while it will not affect the density matrix model. This is because the density matrix model aggregates information from all electronic states at the input of the model and thus sampling will not affect the interaction between electronic states, while the full wavefunction model will have less information when using sampling.

OrbEvo-DM and OrbEvo-WF have 27,977,056 and 26,963,360 parameters, respectively. We optimize the implementation by sharing the radial function computation for different electronic states. We use automatic mixed precision for acceleration.

### 3.2.5 SO(2)-EQUIVARIANT ELECTRIC FIELD CONDITIONING

Following Gupta & Brandstetter (2023); Herde et al. (2024); Helwig et al. (2025), we use FiLM (Perez et al., 2018) like method to insert the conditioning information by computing a scaling factor and a shifting factor from the conditioning, and apply them to the feature map. We apply the conditioning after each layer norm layer in the graph transformer blocks.

Since the feature maps are equivariant features, the conditioning features must also satisfy the equivariant constraints. Specifically, we apply a different scaling factor to different $\ell$'s and compute the bias according the direction of the electric field. In our case, the electric field is always along the $z$-axis, so the spherical harmonics encoding of it is a vector with non-zero entries at $m = 0$ positions and zero otherwise. Mathematically,

$$y_\ell = s_\ell \odot LN(x)_\ell + b_\ell, \tag{9}$$

where $LN(x)_\ell \in \mathbb{R}^{N \times (2\ell+1) \times C}$, $s \in \mathbb{R}^{1 \times 1 \times C}$. $b_\ell \in \mathbb{R}^{1 \times (2\ell+1) \times C}$ is nonzero for $m = 0$ and zero otherwise. Here $LN$ is an SO(3)-equivariant LayerNorm as in EquiformerV2, $s_\ell$ and $b_\ell$ are computed using a MLP from the electric field intensities at current next time bundles, and $\odot$ is multiplication with broadcasting. Since the scale term $s_\ell$ is the same for each $\ell$, it preserves the

SO(3) equivariance. On the other hand, the bias term $b_\ell$ adds predefined directional information into the features and consequently breaks the SO(3) equivariance to SO(2).

We show in the ablation studies in Appendix B that breaking the symmetry is essential to correctly learn the mapping from ground state to the first evolution step. The SO(2)-equivariance of the OrbEvo model is tested in Appendix H.2, Figure 8.

**Wavefunction readout.** We apply an additional equivariant graph attention block to readout the wavefunctions, which is the same as the force prediction in EquiformerV2 but we keep the order up to $\ell = 2$.

## 3.3 TRAINING STRATEGY

**Loss.** We use the per-atom $\ell2$-MAE loss (Chanussot et al., 2021; Liao et al., 2024), defined as

$$\ell_2\text{-MAE}(\Delta^{\text{pred}}, \Delta^{\text{target}}) = \frac{1}{N_a^{\text{batch}}} \sum_{i=1}^{N_a^{\text{batch}}} \|\Delta_{\cdot,i}^{\text{pred}} - \Delta_{\cdot,i}^{\text{target}}\|_2, \quad (10)$$

where $\Delta^{\text{pred}}$ and $\Delta^{\text{target}}$ are the predicted and ground-truth delta wavefunction coefficients (Equation 3), respectively, $\Delta_{\cdot,i}^{\text{pred}}$ and $\Delta_{\cdot,i}^{\text{target}}$ denote the predicted and ground-truth coefficients for the $i$-th atom in the batch, where different orbitals are concatenated into one vector, and $\|\cdot\|_2$ denotes the $\ell_2$-norm. The atom index runs over all sampled electronic states and all molecules in a batch. The loss is averaged over all time steps in the time bundle.

**Push-forward training.** Although training inputs $\Delta(t-h), \ldots, \Delta(t-1)$ are uncorrupted by error, a distribution shift occurs during auto-regressive rollout, where errors made in previous predictions leads to inputs $\Delta(t-h) + \varepsilon(t-h), \ldots, \Delta(t-1) + \varepsilon(t-1)$. Previous works have attempted to mitigate this misalignment by intentionally corrupting training inputs with errors $\hat{\varepsilon}(i)$ sampled from a distribution approximating the rollout error distribution. Pushforward training (Brandstetter et al., 2022) samples these errors directly from the onestep error distribution of the model as

$$\hat{\varepsilon}(t-h), \ldots, \hat{\varepsilon}(t-1) = \text{StopGrad}\left(\mathcal{M}\left(\mathbf{C}(0), \Delta(t-2h : t-h-1)\right) - \Delta(t-h : t-1)\right). \quad (11)$$

Practically, this amounts to letting the model unroll once and then use the unrolled prediction as the new input. However, the onestep error distribution at the outset of training produces noise that dominates the signal at the beginning of training. Thus, in addition to maintaining uncorrupted inputs $\Delta_i$ or adding $\Delta_i + \hat{\varepsilon}_i$ from the pushforward distribution with equal probability, we multiply the $\hat{\varepsilon}_i$ with a warm-up factor which increases linearly from 0 to a maximum value of 1 according to the training step.[1] Finally, because the first target $\Delta(1), \Delta(2), \ldots, \Delta(h)$ cannot be modeled with pushed-forward inputs, we double the weight for its loss in any batch that it appears in to balance its utilization relative to other targets, which can all be modeled using either pushed-forward or uncorrupted targets.

## 4 EXPERIMENTS

### 4.1 DATASET DESCRIPTION

We randomly selected $5,000$ diverse molecules from the QM9 (Ramakrishnan et al., 2014) dataset to demonstrate the generalization capability of our model, and use $1,500$ molecular configurations of the malonaldehyde (MDA) molecule from the MD17 dataset (Chmiela et al., 2018) for the ablation study. Both QM9 and MD17 are widely used in machine learning for materials science and computational chemistry. We then performed self-consistent field (SCF) DFT calculation for each molecule to obtain their ground-state Kohn-Sham wavefunctions using the open-source ABACUS software package (Chen et al., 2010; Li et al., 2016; Lin et al., 2024). Subsequently, we carried out RT-TDDFT calculations to propagate all occupied electronic states for 5 fs in a total of $1,000$ steps with a time step of $0.005$ fs under a spatially uniform, time-dependent electric field. During each time step, wavefunction coefficient matrices were extracted and uniformly downsampled for every 10 steps. After downsampling, each time-dependent wavefunction trajectory contained 101 steps including the first step, which were used as input data for the training, validation, and testing of our OrbEvo model. More details about dataset generation and description can be found in Appendix F.

---

[1] We note that the push-forward warm-up factor may not always be helpful, as shown in Appendix E.

Table 1: Results on the MDA dataset.

| OrbEvo Model | Wavefunction | | | Dipole | | Absorption |
| | 1-step $\ell_2$-MAE | Rollout $\ell_2$-MAE | Rollout nRMSE | nRMSE-all | nRMSE-z | nRMSE-$\alpha$ |
| --- | --- | --- | --- | --- | --- | --- |
| DM-s8 | 0.0242 | 0.0947 | 0.1778 | **0.3008** | **0.2326** | **0.0671** |
| WF-sall | **0.0192** | **0.0853** | **0.1585** | 0.3957 | 0.3066 | 0.0865 |

Table 2: Results on the QM9 dataset.

| OrbEvo Model | Wavefunction | | | Dipole | | Absorption |
| | 1-step $\ell_2$-MAE | Rollout $\ell_2$-MAE | Rollout nRMSE | nRMSE-all | nRMSE-z | nRMSE-$\alpha$ |
| --- | --- | --- | --- | --- | --- | --- |
| DM-s8 | 0.0190 | **0.0797** | **0.1885** | **0.1946** | **0.1459** | **0.0752** |
| WF-sall | **0.0164** | 0.0874 | 0.2071 | 0.6045 | 0.4629 | 0.1270 |

## 4.2 SETUP

**Dataset split and normalization.** For QM9, we use 4,000 molecules for training, 500 molecules for validation, and 500 molecules for testing. For MDA, we use 800 configurations for training, 200 configurations for validation, and 500 configurations for testing. We normalize the initial and delta wavefunction coefficients by dividing their respective orbital-wise rooted mean square (RMS) across all orbitals in training dataset. For delta wavefunction, we also average across time. We normalize the electric field by scaling the maximum intensity to 1.

**Evaluation metrics.** We evaluate the performance of our OrbEvo model on three key physical properties: time-dependent wavefunction coefficients, time-dependent dipole moments, and optical absorption spectra characterized by dipole oscillator strengths. These properties are crucial for downstream tasks in TDDFT, and thus provides a comprehensive evaluation of the model's outputs. The detailed information about these three metrics are provided in Appendix C.

## 4.3 RESULTS

### 4.3.1 QUANTITATIVE RESULTS

The results on MDA and QM9 datasets are summarized in Table 1 and Table 2 respectively. The wavefunction coefficients do not have a unit. The nRMSE errors also do not have units since they are relative errors. Hence all metrics in the tables are unitless.

Overall, the results on the QM9 dataset shown in Table 2 suggest that the OrbEvo-DM model using density matrix as interaction between occupied electronic states outperforms the OrbEvo-WF model which employs layer-wise pooling of the features of occupied electronic states. This may be because the density matrix in the OrbEvo-DM model is inherently consistent with the mathematical formulation of TDDFT: the density functional is used to evaluate the time-dependent Kohn–Sham Hamiltonian in RT-TDDFT. Consequently, it is more straightforward for the OrbEvo-DM model to learn the time evolution operator which depends directly on the density matrix $\boldsymbol{D}(t)$.

We conduct ablation studies on the MDA dataset to verify the model design choices and training strategies. A lower wavefunction error shows a model's ability to evolve the wavefunctions in time while a lower error in dipole and absorption shows a model's ability in capturing the underlying physics. The results are summarized in Appendix B. We also note that the results of OrbEvo-DM-s8 can be further improved with minor changes in training, as shown in Appendix E. Additionally, we report the training and inference cost, as well the simulation time using the classical solver in Appendix D, out-of-distribution analysis in Appendix I, and time bundling analysis in Appendix J.

### 4.3.2 QUALITATIVE RESULTS

We show the computed dipole and absorption spectra produced by OrbEvo-DM-s8 in Figure 3. The plots show that the wavefunctions produced by OrbEvo-DM-s8 starting from ground states can reproduce the per-time-step dipole moment with high correlation. The optical absorption produced by the dipole prediction can faithfully locate the peaks in the spectra, which provides insightful

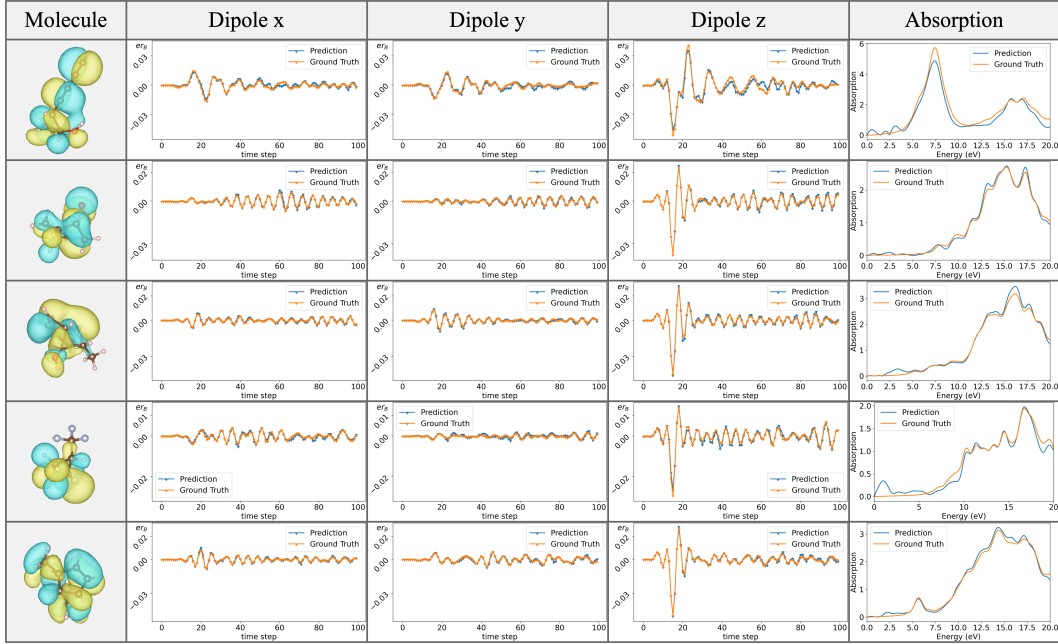

Figure 3: QM9 dipole and absorption with the OrbEvo-DM-s8 model on test samples 0, 10, 20, 30, 40. Note that the test samples are randomly shuffled during dataset generation. The unit for dipole in the plot is $er_B$, where $r_B$ is Bohr radius (0.529 Å). The unit for absorption spectra is $0.529e\text{Å}^2/V$. We highlight that there is no explicit supervision on dipole or absorption during training and validation.

information into the molecular excited states. We also show the wavefunction rollout using OrbEvo-DM-s8 in Figure 4, which demonstrates the good match against the ground truth wavefunctions. Finally, we show plots for MDA dipole and absorption predictions in Appendix E.

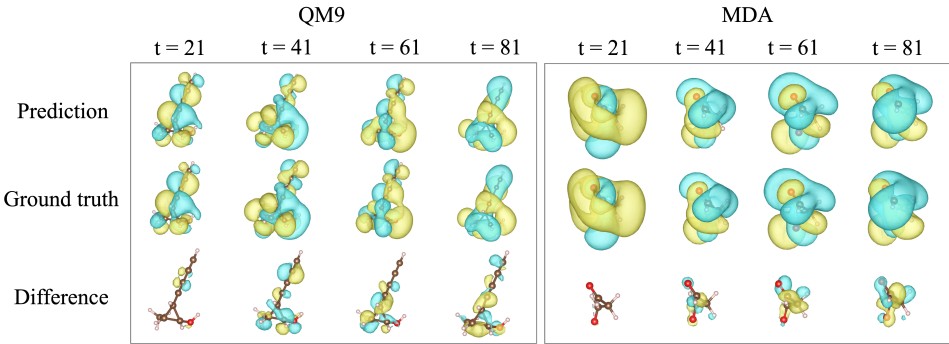

Figure 4: Wavefunction rollout using the OrbEvo-DM-s8 model compared with the ground truth.

## 5 CONCLUSION

In this paper, we propose OrbEvo, which is built upon an equivariant graph transformer architecture. We identify the key issues in modeling inter-electronic-state interaction and propose to model electronic states as separate graphs. We further propose models based on density matrix featurization and full wavefunction pooling interaction. Together with pushforward training, our models can accurately learn the wavefunction evolution accurately. Moreover, we show that the density-matrix-based model is able to learn the underlying physical properties without providing explicit supervising signal to the model. However, standard TDDFT faces limitations, such as difficulties in dealing with conical intersection, and its performance is limited by the accuracy of exchange-correlation energy functionals, which remains an important direction for future development.

ACKNOWLEDGMENTS

This work was supported in part by the U.S. Department of Energy Office of Basic Energy Sciences under grant DE-SC0023866, National Science Foundation under grant MOMS-2331036, National Institutes of Health under grant U01AG070112, and Advanced Research Projects Agency for Health under 1AY1AX000053. We thank Texas A&M HPRC for providing CPU resources.

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

APPENDIX

## A    RELATED WORKS

DFT surrogate models aim to bypass the expensive self-consistency calculation by directly mapping from inputs to the converged DFT outputs. Hamiltonian prediction models (Schütt et al., 2019; Unke et al., 2021; Yu et al., 2023) learn to map from atom types and their 3D coordinates to the converged Hamiltonian matrix. Equivariant 3D graph neural networks enable effective learning with spherical basis through tensor products, albeit the increased computational complexity. For efficiency, eSCN (Passaro & Zitnick, 2023) reduces SO(3) tensor products to SO(2) operations by rotating the relative direction. EquiformerV2 (Liao et al., 2024) incorporates the eSCN convolution into a graph transformer architecture. These models take atom types and coordinates as input. We extend it to a setting where the input features are also high-order equivariant features.

Besides molecules, machine learning has enabled surrogate models for time-dependent PDEs (Li et al., 2021; Tran et al., 2023; Gupta & Brandstetter, 2023; Zhang et al., 2024a) for applications such as modeling fluid dynamics. These surrogate models frequently require conditioning on external information, such as force magnitude or time steps Gupta & Brandstetter (2023); Herde et al. (2024); Helwig et al. (2025). PDE surrogate models have also been developed for graph data (Brandstetter et al., 2022), where the pushforward trick and temporal bundling were proposed to enhance stability over long time-integration periods. We adopt the temporal bundling and apply pushforward training on more realistic 3D graph. While Lippe et al. (2023) showed that the pushforward training may not be helpful in general settings, we show that it can be indeed helpful for realistic graph data.

Machine learning TDDFT is relatively under-explored. Suzuki et al. (2020) use neural networks to improve the exchange-correlation potential in TDDFT. Boyer et al. (2024) learns dipole moments using ridge regression. For time propagation within the ML-PDE paradigm, Shah & Cangi (2024; 2025) study the evolution of charge density in one-dimensional diatomic systems. TDDFT-Net (Zhang et al., 2024b) learns the density evolution starting from the ground-state density for complex molecules. To the best of our knowledge, no existing work directly addresses the learning

Table 3: Ablation studies on the MDA dataset.

| OrbEvo Model | Wavefunction | | | Dipole | | Absorption |
| | 1-step $\ell_2$-MAE | Rollout $\ell_2$-MAE | Rollout nRMSE | nRMSE-all | nRMSE-z | nRMSE-$\alpha$ |
| --- | --- | --- | --- | --- | --- | --- |
| DM-sall | 0.0244 | 0.0997 | 0.1888 | 0.3203 | 0.2494 | 0.0729 |
| DM-s8 | 0.0242 | 0.0947 | 0.1778 | **0.3008** | **0.2326** | **0.0671** |
| DM-s4 | 0.0257 | 0.1010 | 0.1902 | 0.3096 | 0.2396 | 0.0734 |
| DM-sall-cat | 0.1269 | 0.4429 | 0.7875 | 2.063 | 1.6345 | 0.8040 |
| DM-s8-no-dm(t) | 0.0508 | 0.2788 | 0.5457 | 0.8738 | 0.6768 | 0.1758 |
| DM-s8-onestep | 0.0200 | 0.1501 | 0.2851 | 0.4369 | 0.3386 | 0.1211 |
| WF-sall | **0.0192** | **0.0853** | **0.1585** | 0.3957 | 0.3066 | 0.0865 |
| WF-s8 | 0.0334 | 0.2074 | 0.4054 | 0.6579 | 0.5218 | 0.1338 |
| WF-s4 | 0.0414 | 0.2527 | 0.4961 | 0.7762 | 0.6104 | 0.1582 |
| WF-sall-onestep | 0.0205 | 0.1978 | 0.3708 | 0.7400 | 0.5754 | 0.1590 |
| WF-sall-inv-cond | 0.0224 | 0.6773 | 1.1564 | 1.3405 | 1.2632 | 0.1667 |

of time-dependent wavefunctions, representing a critical gap in the field. Here we study TDDFT directly in the wavefunction space, which captures the underlying physical process and enables more accurate predictions. The orbital-based representation that we adopted also allows for more efficient data encoding.

# B  ABLATION STUDIES

We conduct ablation studies on the MDA dataset to verify the model design choices and training strategies. A lower wavefunction error shows a model's ability to evolve the wavefunctions in time while a lower error in dipole and absorption shows a model's ability in capturing the underlying physics. The results are summarized in Table 3.

**Electronic states sampling.** Models with suffix "-all" use all electronic states during training. Models end with "-s8" and "-s4" randomly sample 8 and 4 electronic states during training, respectively. The results show that the sampling does not affect OrbEvo-DM's performance while it degrades the performance of OrbEvo-WF significantly. It shows that by aggregating the electronics state information early via density matrix can effectively capture the inter-electronic-state interaction. The OrbEvo-WF results show the importance of considering all electronic states' information.

**Electronic state graph construction.** In DM-sall-cat, we concatenate wavefunctions from all electronic states along the channel dimension at model's input instead of considering them as individual graphs. The result shows that the model cannot learn the wavefunction mapping correctly, demonstrating the importance of our graph modeling method.

**Density matrix ablation.** In DM-s8-no-dm(t), we remove the dependency on the time-evolving density matrix. The results show that the model cannot learn correctly, showing the importance of time-evolving density in learning the propagation.

**Training strategy.** We show the results without using pushforward for DM-s8-onestep and WF-sall-onestep. The results show that although the models are able to learn the one-step mapping more accurately, the rollout error is significantly worse, showing importance of pushforward training for learning error accumulation during rollout.

**Equivariant conditioning.** In WF-sall-inv-cond, we disable the equivariant electric field conditioning and add the bias term into the invariant $\ell = 0$ part instead. The results show that although the one-step error can go down normally, the rollout does not work. We observe that the model cannot learn the mapping from the initial ground state to the first step correctly, although it is able to evolve the subsequent steps given the ground truth.

## C    EVALUATION METRICS

### C.1    DENSITY CONSERVATION

The propagation of electronic wavefunctions conserves the total density. The square of electron density for electronic state $n$ is computed as

$$\mathbf{C}_n(t)^\dagger \boldsymbol{S} \mathbf{C}_n(t) \in \mathbb{R}, \tag{12}$$

which is equal to 1, where $\boldsymbol{S}$ is the overlap matrix.

We normalize the predicted coefficients $\mathbf{C}_n(t)$ as

$$\widetilde{\mathbf{C}}_n(t) = \frac{\mathbf{C}_n(t)}{\sqrt{\mathbf{C}_n^\dagger(t)\boldsymbol{S}\mathbf{C}_n(t)}} \tag{13}$$

We note that the global phase $\gamma_n(t)$ in Equation 3 cancels out in the product.

### C.2    WAVEFUNCTION METRIC

We report the $\ell_2$-MAE error (Equation (10)) for the time-dependent wavefunctions. For a more interpretable metric, we also report the normalized rooted mean square (nRMSE) error, defined for each molecule as

$$\mathrm{nRMSE}(\mathbf{C}^{\mathrm{pred}}, \mathbf{C}^{\mathrm{target}}) = \frac{\sum_{n=1}^{N_{\mathrm{occ}}} \sqrt{\sum_{t=1}^{T} \sum_{o=1}^{N_{\mathrm{orb}}} \|\mathbf{C}_{t,n,o}^{\mathrm{pred}} - \mathbf{C}_{t,n,o}^{\mathrm{target}}\|_2^2}}{\sum_{n=1}^{N_{\mathrm{occ}}} \sqrt{\sum_{t=1}^{T} \sum_{o=1}^{N_{\mathrm{orb}}} \|\mathbf{C}_{t,n,o}^{\mathrm{target}}\|_2^2}}, \tag{14}$$

where $N_{\mathrm{occ}}$ and $N_{\mathrm{orb}}$ denote the number of occupied electronic states and local atomic orbital bases in the molecule.

### C.3    DIPOLE MOMENT

Dipole moment describes the density distribution over spatial directions and are defined as $\langle\psi|\hat{\boldsymbol{r}}_m|\psi\rangle$, where $\hat{\boldsymbol{r}}_m$ is the position operator along $m \in \{x, y, z\}$ direction. With the local atomic orbital basis, given the position matrices for three spatial directions $\mathfrak{r}_{m,ij} = \langle\phi_i|\hat{\boldsymbol{r}}_m|\phi_j\rangle \in \mathbb{R}^{N_{\mathrm{orb}} \times N_{\mathrm{orb}}}$, the dipole moment of each molecule can be computed as

$$\mathbf{p}_m(t) = \sum_{n=1}^{N_{\mathrm{occ}}} \eta_n \tilde{\mathbf{C}}_n(t)^\dagger \mathfrak{r}_m \tilde{\mathbf{C}}_n(t), \quad m \in \{x, y, z\}, \tag{15}$$

where density conservation is applied to the unrolled wavefunctions as a post-processing step prior to computing the dipoles. We are interested in the dipole difference against time 0: $\Delta\mathbf{p}_m(t) = \mathbf{p}_m(t) - \mathbf{p}_m(0), \ m \in \{x, y, z\}$. We report the nRMSE of the dipole moment for all directions, defined as

$$\mathrm{nRMSE\text{-}all}\left(\Delta\mathbf{p}^{\mathrm{pred}}, \Delta\mathbf{p}^{\mathrm{target}}\right) = \frac{\sqrt{\sum_{t=1}^{T} \sum_{m \in \{x,y,z\}} \left(\Delta\mathbf{p}_m^{\mathrm{pred}}(t) - \Delta\mathbf{p}_m^{\mathrm{target}}(t)\right)^2}}{\sqrt{\sum_{t=1}^{T} \sum_{m \in \{x,y,z\}} \left(\Delta\mathbf{p}_m^{\mathrm{target}}(t)\right)^2}}, \tag{16}$$

as well as for $z$ direction, defined as

$$\mathrm{nRMSE\text{-}z}\left(\Delta\mathbf{p}^{\mathrm{pred}}, \Delta\mathbf{p}^{\mathrm{target}}\right) = \frac{\sqrt{\sum_{t=1}^{T} \left(\Delta\mathbf{p}_z^{\mathrm{pred}}(t) - \Delta\mathbf{p}_z^{\mathrm{target}}(t)\right)^2}}{\sqrt{\sum_{t=1}^{T} \left(\Delta\mathbf{p}_z^{\mathrm{target}}(t)\right)^2}}. \tag{17}$$

## C.4 Optical Absorption

Optical absorption is an important physical property which reflects the ability of molecule to absorb light at specific frequencies. It is characterized by dipole oscillator strength which can be calculated from the time-dependent dipole moment in response to the applied external electric field as follows:

$$\alpha_z(\omega) = \text{Im}\left[\frac{\int \mathbf{p}_z(t)e^{i\omega t}dt}{\int E_z(t)e^{i\omega t}dt}\right]. \tag{18}$$

We report the nRMSE for the dipole oscillator strength along the $z$ direction, defined as

$$\text{nRMSE-}\alpha\left(\alpha_z^{\text{pred}}, \alpha_z^{\text{target}}\right) = \frac{\sqrt{\sum_\omega \left(\alpha_z^{\text{pred}}(t) - \alpha_z^{\text{target}}(t)\right)^2}}{\sqrt{\sum_\omega \left(\alpha_z^{\text{target}}(\omega)\right)^2}}. \tag{19}$$

## D Computational Cost & Comparison

In this section we report the training (Table 4) and inference cost of OrbEvo (Table 5). We also report the simulation time with the classical solver ABACUS (Table 6).

| Dataset | Model | # iterations | GPU | Wall Clock Time | GPU Memory (MB) |
|---|---|---|---|---|---|
| MDA | OrbEvo-DM-s8 | 300k | 2× 11GB 2080Ti | 3.475 days | 13,848 |
| MDA | OrbEvo-WF | 300k | 2× 11GB 2080Ti | 3.345 days | 14,248 |
| QM9 | OrbEvo-DM-s8 | 395k | 4 × 48GB A6000 | 3.118 days | 49,434 - 54,700 |
| QM9 | OrbEvo-WF | 395k | 2 × 80GB A100 | 5.003 days | 46,652 - 69,662 |

Table 4: Training cost of OrbEvo models. MDA models are trained with a batch size 32. QM9 models use a batch size of 16. All models are trained with Pytorch distributed data parallel (`torch.ddp`) for multi-gpu training and with `num_workers=16` in dataloader for MDA and `num_workers=32` for QM9. As a rough estimation, 2× 2080Ti is roughly equivalent to 1×A6000 in terms of speed. The GPU memory usage is tested by running training on 1 single A100 GPU for 10 minutes. For QM9, The GPU memory can vary depending on the molecule sizes in a batch. We note that with a slightly optimized push-forward training implementation, we are able to fit the training within 2×A6000 GPUs for both OrbEvo models on the QM9 dataset with similar training time.

| Dataset | Model | GPU | Batch Size | Wall Clock Time / Batch | | GPU Memory (MB) |
|---|---|---|---|---|---|---|
| | | | | Wavefunction | Wavefunction + Property | |
| MDA | OrbEvo-DM | 1× A6000 | 20 | 3.67 seconds | 5.23 seconds | 5742 |
| MDA | OrbEvo-WF | 1× A6000 | 20 | 2.84 seconds | 4.60 seconds | 2032 |
| QM9 | OrbEvo-DM | 1× A6000 | 20 | 18.00 seconds | 26.74 seconds | 34,164 - 42,842 |
| QM9 | OrbEvo-WF | 1× A6000 | 20 | 11.86 seconds | 20.31 seconds | 17,204 |

Table 5: Inference cost of OrbEvo models. All models are tested one a single A6000 GPU using `num_workers=10` in dataloader. The reported times are wall clock time per batch. We report both the time for producing the wavefunction trajectory (Wavefunction), as well as the time for producing the wavefunction trajectories and computing the dipoles and absorptions (Wavefunction + Property). Note that the properties are not parallelized with batch processing and are computed on CPUs. We note that electronic state sampling is not enabled during inference, which leads to increased GPU memory usage for OrbEvo-DM. In comparison, during training OrbEvo-DM is able to use electronic state sampling to reduce GPU usage.

## E Qualitative Results on MDA and Efficient Training

We show the dipole and absorption produced using the predicted wavefunctions on MDA samples in Figure 5, where we train the OrbEvo-DM-s8 model using push-forward training with some minor

| Dataset | # CPU cores | Wall Clock Time / Molecule Ground-state DFT | Total |
|---------|-------------|---------------------------------------------|-------|
| MDA | 24 | 34.3 seconds | 1.5 hours |
| QM9 | 24 | 73.1 seconds | 3.2 hours |

Table 6: Simulation time per molecule. The simulation time is averaged over 40 simulations. Ground-state DFT is the time to compute the initial wavefunction coefficients from molecular structures. The initial wavefunction coefficients are used as input to OrbEvo models.

changes compared to the models in the main text: (1) We disable the linear warm-up factor and always enable pushforward with a 50% probability. (2) We switch the model to evaluation mode during push-forward unrolling. This could make the push-forward noise closer to the real rollout during test. (3) We fix the number of push-forward samples to be half of the per-GPU batch size to have a more stable GPU usage. The result of models trained in this way is summarized in Table 7. On the other hand, we observe that such changes in training may not be helpful for OrbEvo-WF.

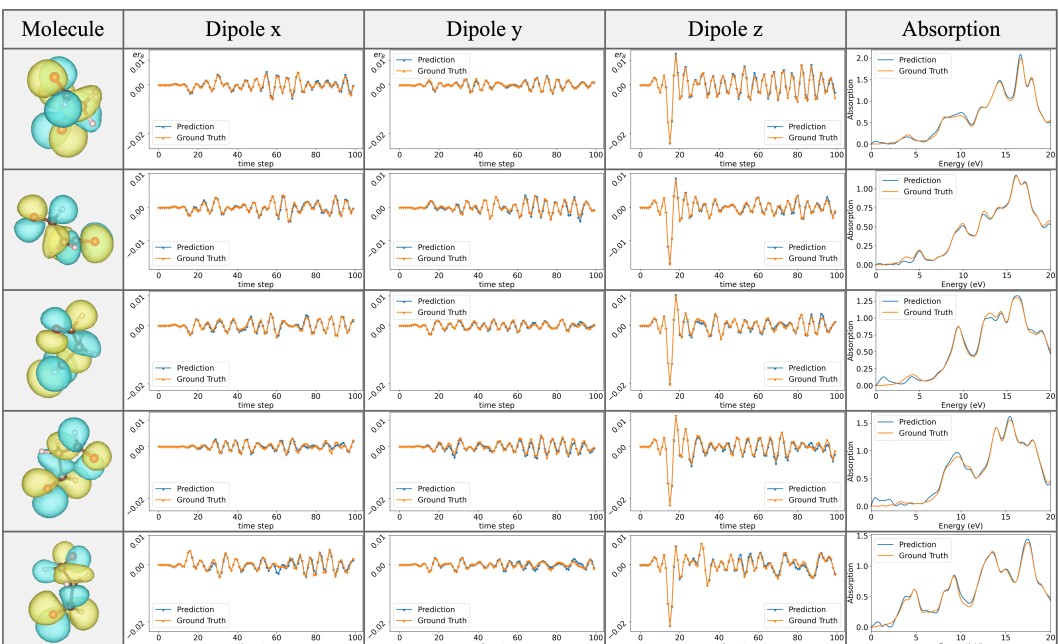

Figure 5: MDA dipole and absorption with the OrbEvo-DM-s8 model on test samples. The unit for dipole in the plot is $er_B$, where $r_B$ is Bohr radius (0.529 Å). The unit for absorption spectra is $0.529e\text{Å}^2/V$.

Table 7: Results on the MDA dataset with the new training.

| OrbEvo Model | Wavefunction | | | Dipole | | Absorption |
|--------------|--------------|--------|--------|--------|--------|------------|
| | 1-step $\ell_2$-MAE | Rollout $\ell_2$-MAE | Rollout nRMSE | nRMSE-all | nRMSE-z | nRMSE-$\alpha$ |
| DM-s8 | **0.0224** | **0.0863** | **0.1613** | **0.1997** | **0.1499** | **0.0539** |
| WF-sall | 0.0225 | 0.1080 | 0.2008 | 0.3758 | 0.2881 | 0.0822 |

## F  DATASET DESCRIPTION

The molecules and their configurations used in this work were sourced from the QM9 (Ramakrishnan et al., 2014) and MD17 databases (Chmiela et al., 2018). The QM9 dataset contains a large number of chemically diverse molecules. This combination allows our model to cover a wide range

of potential molecular behaviors and properties. The MD17 dataset provides high-resolution molecular dynamics trajectories for a small number of molecules with many different conformations. Both QM9 and MD17 are widely used in machine learning for materials science and computational chemistry. For this work, we randomly chose $5,000$ different molecules from the QM9 dataset consisting of C, H, O, and N elements to demonstrate the generalization capability of our model, and randomly selected $1,500$ molecular configurations of the malonaldehyde (MDA) molecule from the MD17 dataset for the ablation study.

To generate the RT-TDDFT datasets for the above QM9 and MDA molecules, we utilized the open-source ABACUS software package (Chen et al., 2010; Li et al., 2016; Lin et al., 2024) to perform the DFT and RT-TDDFT calculations. Consistent input parameters were used to ensure comparability between datasets. Specifically we employed the SG15 Optimized Norm-Conserving Vanderbilt (ONCV) pseudopotentials (SG15-V1.0) (Hamann, 2013), a standard atomic orbitals basis set hierarchically optimized for the SG15-V1.0 pseudopotentials (Lin et al., 2021), and a kinetic energy cutoff of 100 Rydberg. The ground-state Kohn-Sham wavefunctions were obtained by self-consistent field (SCF) calculations of DFT with a dimensionless convergence threshold of $10^{-6}$.

For RT-TDDFT calculations, we used ground-state Kohn-Sham wavefunctions as the initial states at $t = 0$ and performed time propagation for 5 fs in a total of $1,000$ steps with a time step of $0.005$ fs. To simulate the quantum dynamics of the system under an external field, a time-dependent uniform electric field $E_z(t)$ was applied along the $z$ direction:

$$E_z(t) = E_0 \left(\cos[2\pi f_1(t - t_0)] + \cos[2\pi f_2(t - t_0)]\right) \exp\left[-\frac{(t - t_0)^2}{2\sigma^2}\right].$$

It consists of two frequencies of $f_1 = 3.66$ fs$^{-1}$ and $f_2 = 1.22$ fs$^{-1}$, with a Gaussian width $\sigma = 0.2$ fs, a field amplitude $E_0 = 0.01$ V/Å, and a central time of $t_0 = 0.75$ fs. During each time step, wavefunction coefficient matrices were saved and then extracted, serving as input data for our model training, validation and testing.

To enhance computational efficiency and accuracy, we modified the ABACUS source code to calculate the overlap matrix only once at $t = 0$. Furthermore, we ensured that the output matrix retained 16 significant digits of precision. This modification allowed us to generate reliable data with greater efficiency, making it well suited for model training and testing. The DFT and RT-TDDFT calculations were performed using 24 parallel CPU cores.

## G  TIME EVOLUTION OF KOHN-SHAM WAVEFUNCTIONS IN RT-TDDFT

In RT-TDDFT, each Kohn-Sham wavefunction $\psi_i$ evolves in time under the time-ordered evolution operator $\hat{U}(t, t_0)$, starting from the initial time $t_0$: $\psi_i(t) = \hat{U}(t, t_0)\psi_i(t_0)$, where

$$\hat{U}(t, t_0) = \hat{\mathcal{T}}\exp\left(-\frac{i}{\hbar}\boldsymbol{S}^{-1}\int_{t_0}^{t}\hat{\boldsymbol{H}}(t')dt'\right).$$

$\hat{\mathcal{T}}$ is time-ordering operator. In RT-TDDFT, total simulation time $T_{\text{tot}}$ is discretized into $N_{\text{tot}}$ steps with each time step of $\Delta t = T_{\text{tot}}/N_{\text{tot}}$, and $\hat{U}(t, t_0)$ is approximated by the product of evolution operators over the discretized time grid (Gómez Pueyo et al., 2018),

$$\hat{U}(t, t_0) = \prod_{m=1}^{N_{\text{tot}}} \hat{U}[t_0 + m\Delta t, t_0 + (m - 1)\Delta t].$$

In general, $\hat{U}[t_0 + m\Delta t, t_0 + (m-1)\Delta t]$ should satisfy the unitary condition to conserve the density: $\hat{U}^{\dagger}[t_0 + m\Delta t, t_0 + (m - 1)\Delta t] = \hat{U}^{-1}[t_0 + m\Delta t, t_0 + (m - 1)\Delta t]$. Moreover, for molecules and solids under external electric field, it should satisfy time-reversal symmetry: $\hat{U}[t_0 + m\Delta t, t_0 + (m - 1)\Delta t] = \hat{U}[t_0 + (m - 1)\Delta t, t_0 + m\Delta t]$. Such time evolution needs to be applied to all occupied electronic states for $N_t$ time steps, making it computationally demanding.

Table 8: Performance on QM9_ood validation and test sets.

| Model | Dataset | 1-step $\ell_2$-MAE | Rollout nRMSE | Dipole $z$ nRMSE | Absorption nRMSE |
|---|---|---|---|---|---|
| OrbEvo-DM-s8-ood | QM9_ood - val | 0.0142 | 0.1498 | 0.1113 | 0.0615 |
| OrbEvo-DM-s8-ood | QM9_ood - test | 0.0132 | 0.1482 | 0.1175 | 0.0697 |

# H  IMPLEMENTATION DETAILS

## H.1  TENSOR PRODUCT

In Figure 6, we visualize the tensor product for computing density matrix feature from wavefunction, which is implemented using `e3nn.o3.FullTensorProduct`.

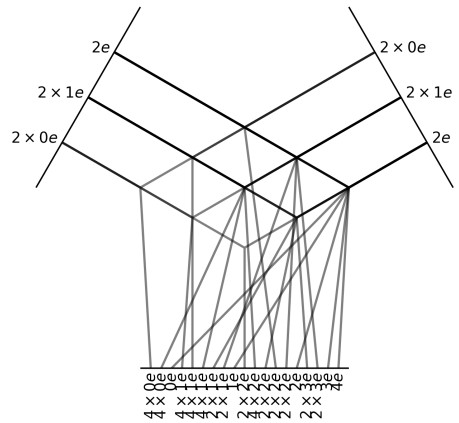

Figure 6: Tensor product visualization produced by the `e3nn` library.

## H.2  EQUIVARIANCE TEST

In this section we test the SO(2)-equivariance error for both the TDDFT numerical simulation and the OrbEvo model.

In Figure 7, we run two simulations using ABACUS with original or rotated molecule. In Figure 8, we use the model to make predictions using inputs before and after rotation. In both cases we rotate around the electric field direction by 35 degree and we conduct manual rotation-transform to align the resulting coefficients or to produce rotation-transformed input. When applying the rotation transformation to the coefficients, s orbitals and $m = 0$ components in p and d orbitals remain unchanged, $m = \pm 1$ components in p and d orbitals are rotated by 35 degree around the electric field direction, and $m = \pm 2$ components in d orbitals are rotated by 70 degree around the electric field direction.

# I  OUT-OF-DISTRIBUTION ANALYSIS

Using our generated 5000 QM9 data, we created an OOD split for it based on the number of atoms in a molecule. In particular, we use the number of atoms from 8 to 20 for training (3955 samples), number of atoms = 21 for validation (518 samples), and number of atoms from 23 to 29 for testing (527 samples). We train the OrbEvo-DM-s8 model using this OOD split, the trained model is dubbed OrbEvo-DM-s8-ood. In the Table 8, we evaluate the trained model on the validation and test sets of the OOD split.

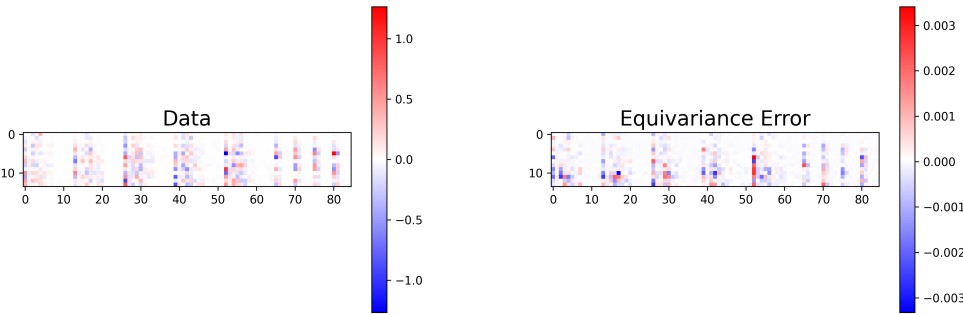

Figure 7: Equivariance error of TDDFT data. Left: real part of the wavefunction coefficients of an unrotated MDA molecule at one time step. Right: the difference between the wavefunctions at the same time step in a second simulation produced from a rotated version of the same molecule, and the coefficients manually rotation-transformed from the left plot. In the second simulation the molecule is rotated by 35 degree around the electric field direction.

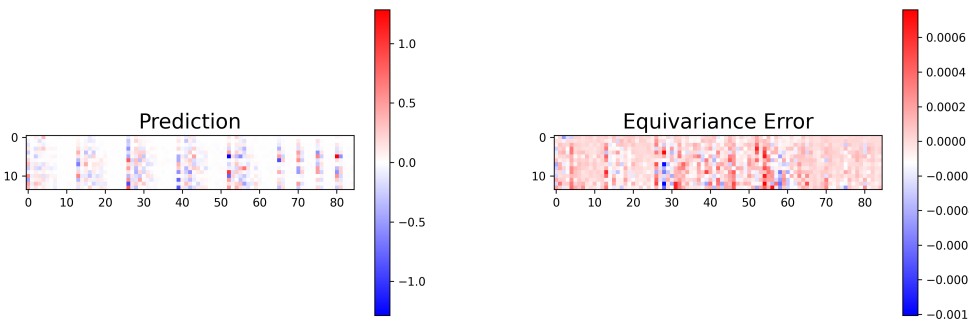

Figure 8: Equivariance error of OrbEvo-DM. Left: real part of the model's predicted wavefunction coefficients for a MDA molecule using the ground-truth wavefunctions at one time step as input. Right: the difference between the model's predicted wavefunctions using the rotated structure and manually rotation-transformed ground-truth wavefunctions, and the coefficients manually rotation-transformed from the left plot. The molecule's rotation and the rotation transformation is 35 degree around the electric field direction.

Rather surprisingly, the OOD split gives better average test accuracy than the random split reported in our main paper (Rollout nRMSE = 0.1885, Dipole z nRMSE = 0.1459, Absorption nRMSE = 0.0752). However, the numbers are not directly comparable since different test data are used. To fairly compare the ID and OOD performance, we evaluate the models on the validation and test data that are both in the OOD split and the random split. Specifically, there are 67 molecules in the intersection of the OOD validation set and the random validation set (QM9_id_ood_intersect - val), and 43 molecules in the intersection of the OOD test set and random test set (QM9_id_ood_intersect - test). We report the results of both the model trained on the OOD split (OrbEvo-DM-s8-ood) and the model trained using the random split (OrbEvo-DM-s8 in the main paper) in Table 9.

Despite the randomness due to the small amount of common validation / test data, we can see that overall the model trained on the random split performs closer to the model trained on the OOD split. Overall, the above results show that the OrbEvo model is able to generalize on larger systems than those in the training data. The results also suggest that larger systems may not necessarily be more challenging for learning the dynamics of wavefunctions, potentially due to the fact that smaller systems can exhibit dynamic patterns with a larger magnitude or more complex behaviors.

Table 9: Performance comparison on QM9_id_ood_intersect validation and test sets.

| Model | Dataset | 1-step $\ell_2$-MAE | Rollout nRMSE | Dipole $z$ nRMSE | Absorption nRMSE |
|---|---|---|---|---|---|
| OrbEvo-DM-s8-ood | QM9_id_ood_intersect - val | **0.0141** | **0.1500** | **0.1126** | 0.0633 |
| OrbEvo-DM-s8 | QM9_id_ood_intersect - val | 0.0146 | 0.1579 | 0.1153 | **0.0628** |
| OrbEvo-DM-s8-ood | QM9_id_ood_intersect - test | **0.0137** | **0.1496** | 0.1142 | 0.0666 |
| OrbEvo-DM-s8 | QM9_id_ood_intersect - test | 0.0139 | 0.1521 | **0.1074** | **0.0636** |

## J    TIME BUNDLING ANALYSIS

We additionally conduct an experiment on the MDA dataset with various time bundle sizes 1, 2, 4, 16 and with 8 (used in the main paper) where a time bundle size 1 corresponds to without time bundling. We report 1-step error (average error for the time bundle), rollout errors of different trajectory lengths (start from time 0, produce trajectories with lengths 8, 16, 32, 64, 100, larger bundle needs less autoregressive steps), as well as relative errors for dipole- and absorption on the full rollout. The test results are summarized in Table 10.

Table 10: Time bundling analysis on the MDA dataset.

| Time bundle size | 1-step $\ell_2$-MAE | 8-step Rollout nRMSE | 16-step Rollout nRMSE | 32-step Rollout nRMSE | 64-step Rollout nRMSE | 100-step Rollout nRMSE | Dipole $z$ nRMSE | Absorption nRMSE |
|---|---|---|---|---|---|---|---|---|
| 1 | **0.0093** | 0.0780 | 0.0340 | 0.1363 | 0.4433 | 0.9032 | 0.9526 | 0.1684 |
| 2 | 0.0130 | **0.0668** | 0.0340 | 0.1106 | 0.3087 | 0.5765 | 0.5669 | 0.1228 |
| 4 | 0.0139 | 0.0693 | **0.0289** | **0.0572** | **0.1235** | 0.1979 | 0.2670 | 0.0758 |
| 8 | 0.0242 | 0.0720 | 0.0378 | 0.0677 | 0.1245 | **0.1778** | **0.2328** | **0.0672** |
| 16 | 0.0588 | 0.1026 | 0.0544 | 0.1075 | 0.2040 | 0.2872 | 0.2641 | 0.0922 |

We observe that using a smaller time bundle size results in a smaller 1-step error. This is because the model needs to predict fewer steps, and closer steps in time are easier to predict. For rollout errors, time bundle sizes of 1 and 2 can produce correlated rollouts at 16 steps, but start to diverge for 32 or more steps. Time bundle size of 4 performs well for 32 steps, but becomes less effective than time bundle size 8 for longer rollout. Time bundle size 8 produces the best wavefunction in the overall rollout. Time bundle size of 16 stays stable but the accuracy is not as good as 8 or 4.

In terms of training, we observe that time bundle size 1 and 2 start to overfit to onestep mapping and the validation rollout errors start to overfit during the training, with the best validation 100-step rollout error of around 0.4 occurs at around 100k iteration for time bundle size 1, and around 0.6 at around 120k iterations for time bundle size 2 (total training iteration is 300k). Moreover, we observe some oscillations in the validation rollout curves for time bundle size 4 during training.

Note that during training we randomly sample from all 100 steps for feasible starting time for onestep training. So the total number of training pairs are similar for different time bundle sizes. Overall, a time bundle size of 8 remains a reasonable choice, in which case the model needs to unroll 13 steps to produce the entire 100-step trajectory.

## K    PREDICTION OF GLOBAL PHASE

We re-purpose the OrbEvo-DM-s8 model to predict the global phase $\gamma(t)$. In particular, we replace the wavefunction readout block with a feedforward network. During training, the model takes the ground truth wavefunction coefficients and the global phases in the current time bundle as input, and predicts the global phases at the next time bundle. The real and imaginary parts are predicted as a 2D vector for each electronic state. We use the MAE as loss function. For MDA, the model is first trained for 100k iterations and subsequently resumed training for 200k iterations. For QM9, the model is first trained for 140k iterations and subsequently resumed training for 186k iterations. We show examples of predicted rollout for QM9 in Figure 9, and for MDA in Figure 10. We observe that the predicted global phases are in good agreement with the ground truth.

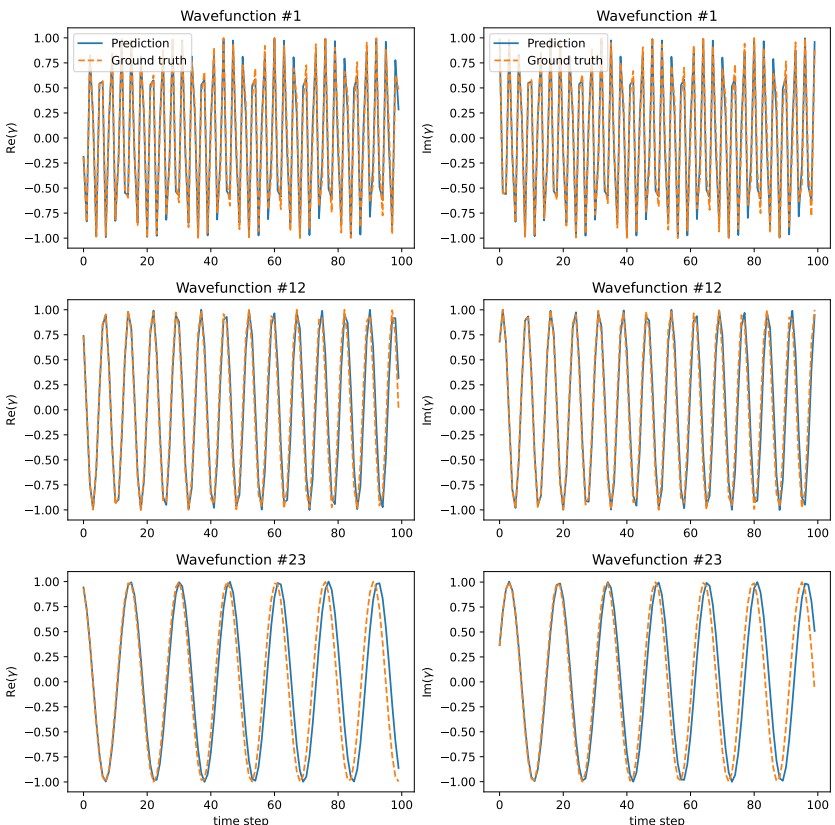

Figure 9: Global phase rollout on the QM9 sample.

## L    ADDITIONAL ABLATIONS

We study the effect of keeping the quadratic term of delta wavefunctions in the density matrix calculation in Table 11, as well as the effect of replacing push-forward training with noise injection in Table 12.

Table 11: Density matrix analysis on the MDA dataset.

| OrbEvo Model | Wavefunction | | | Dipole | | Absorption |
|---|---|---|---|---|---|---|
| | 1-step $\ell_2$-MAE | Rollout $\ell_2$-MAE | Rollout nRMSE | nRMSE-all | nRMSE-z | nRMSE-$\alpha$ |
| DM-s8 | **0.0242** | **0.0947** | **0.1778** | **0.3012** | **0.2329** | **0.0672** |
| DM-s8-w/-quadratic-dm | 0.0290 | 0.1110 | 0.2088 | 0.3538 | 0.2744 | 0.0784 |

Table 12: Noise injection results on the MDA dataset.

| OrbEvo Model | Wavefunction | | | Dipole | | Absorption |
|---|---|---|---|---|---|---|
| | 1-step $\ell_2$-MAE | Rollout $\ell_2$-MAE | Rollout nRMSE | nRMSE-all | nRMSE-z | nRMSE-$\alpha$ |
| DM-s8-noise | 0.0204 | 0.1262 | 0.2423 | **0.3868** | **0.3036** | 0.0815 |
| WF-sall-noise | **0.0155** | **0.0866** | **0.1617** | 0.4045 | 0.3157 | **0.0788** |

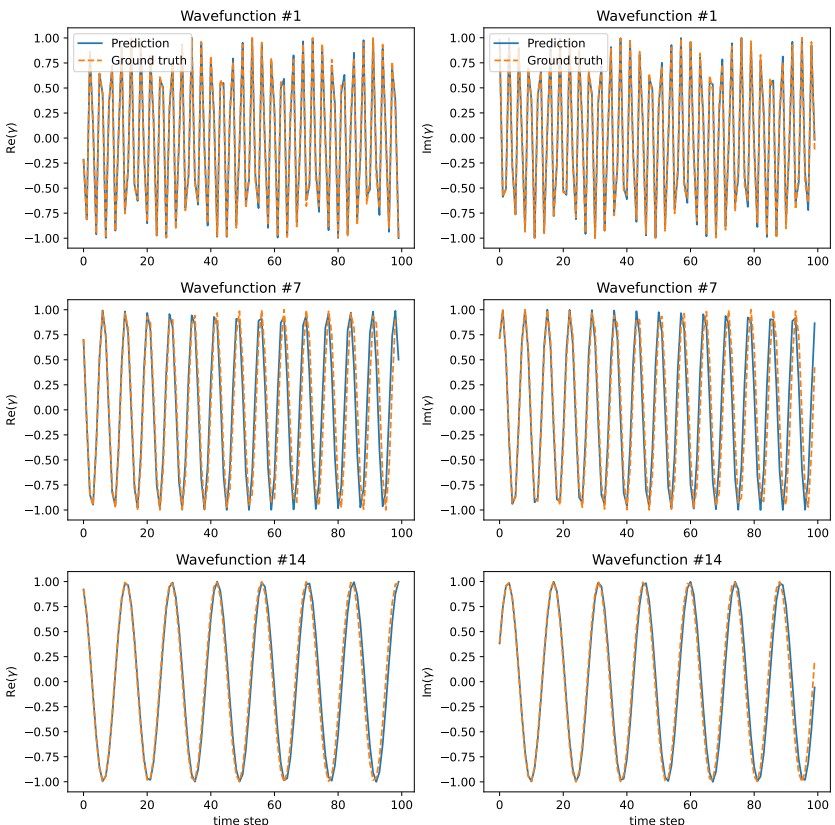

Figure 10: Global phase rollout on the MDA sample.

## M  MODEL HYPERPARAMETERS

We summarize OrbEvo's hyperparameters in Table 13. Most of them are hyperparameters for the EquiformerV2 (Liao et al., 2024) backbone.

## N  LARGE LANGUAGE MODEL USAGE

We use large language models to aid or polish writing sparsely. LLMs are also used lightly to help write data processing scripts.

| Hyperparameters | Value |
|---|---|
| Optimizer | AdamW |
| Learning rate scheduling | Cosine Annealing |
| Maximum learning rate | $1 \times 10^{-3}$ |
| Weight decay | $1 \times 10^{-3}$ |
| Number of epochs for MDA | 129 (300k iterations) |
| Number of epochs for QM9 | 17 (395k iterations) |
| Maximum cutoff radius | 5.0 |
| Number of layers | 6 |
| Number of sphere channels | 128 |
| Number of attention hidden channels | 128 |
| Number of attention heads | 8 |
| Number of attention alpha channels | 32 |
| Number of attention value channels | 16 |
| Number of FFN hidden channels | 512 |
| $\ell_{\max}$ list | [4], [2] |
| $m_{\max}$ list | [4], [2] |
| Grid resolution | eSCN default |
| Number of sphere samples | 128 |
| Number of edge channels | 128 |
| Number of distance basis | 250 |
| Alpha drop rate | 0.1 |
| Drop path rate | 0.05 |
| Projection drop rate | 0.0 |
| Number of future time steps | 8 |
| Number of conditioning time steps | 8 |

Table 13: OrbEvo model hyperparameters.

