# OpenReview forum: "Orbital Transformers for Predicting Wavefunctions in Time-Dependent Density Functional Theory"
_ICLR.cc/2026/Conference — ICLR 2026 Poster_

### Official Review · Reviewer_ce9Z · 2025-10-28

**Soundness:** 3
**Presentation:** 3
**Contribution:** 4
**Rating:** 6
**Confidence:** 4

**Summary:**

This paper tackles the important and challenging problem of accelerating Real-Time TDDFT (RT-TDDFT) computations using deep learning.
Specifically, it adopts an autoregressive framework to accelerate the propagations of RT-TDDFT, where the wavefunctions of previous steps are input into the network for the prediction of the next steps' wavefunctions. The paper proposes two model architectures (OrbEvo-FullWF and OrbEvo-DM) with different electronic state interacting strategies and compares their performance on their self-generated TDDFT dataset.

I think the paper is in a good shape, with nontrivial contributions for a novel application (RT-TDDFT) and specifically designed models (OrbEvo-FullWF and OrbEvo-DM). Nonetheless, there exist several concerns, which should be addressed before acceptance.

**Strengths:**

* Machine learning TDDFT is an important and relatively under-explored research field. This paper approaches the problem with a novel setting (directly learning the wavefunctions), representing a pioneering attempt in this direction. The handling of the high-dimensional orbital coefficient object (the time, the electronic state and the atomic orbital dimensions are all different from the number of atoms) is especially interesting.
* The design choices of the prediction target and the model architectures seem to be thoroughly considered and physics-grounded.
* The practices of improving model performance, such as the delta transformation of wavefunctions, the time bundling setting and the push-forward training, are detailed and provide a good impression of adopting state-of-the-art techniques in related fields.
* The paper is written in a clear and easy-to-understand language, with moderate background knowledge included.

**Weaknesses:**

* Although the paper has provided some background information for RT-TDDFT, I find myself unclear about the big picture and lost in the implementation details. It is not clear how the learned model is used in the RT-TDDFT framework.
* The units are missing from all the performance numbers in the paper.
* Neither the source code nor the dataset are provided. This renders the paper hard to follow.

**Questions:**

* About motivation: Could you provide a clearer picture of the framework? For example, how is the learned model used in RT-TDDFT to accelerate the computation?
* About performance: How much acceleration is achieved? How should one interpret the metrics in Table 1 and Table 2?
* How does the model compare to direct property prediction methods?

---

> ### Author Response · Authors · 2025-11-27
>
> Thank you for taking the time to read and evaluate our paper,  and thank you for the patience in waiting for our response. We reply to your comments and questions in the following.
>
> **W1**. Although the paper has provided some background information for RT-TDDFT, I find myself unclear about the big picture and lost in the implementation details. It is not clear how the learned model is used in the RT-TDDFT framework.
>
> The learned model is used to replace RT-TDDFT’s time-consuming propagation solving the TDDFT equation. Given a molecule’s 3D structure and a time-dependent electric field excitation, the procedure of RT-TDDFT simulation is as follows. We first compute the ground state wavefunctions using ground-state DFT. We then solve the time propagation and produce the wavefunctions at each later time step. In this work, our learned model targets the time-dependent propagation part. So the RT-TDDFT framework with the learned model will first compute the ground state wavefunctions using ground-state DFT calculation, and then apply the learned propagation model autoregressively to produce the wavefunctions in the later time without computationally expensive explicit time integration used in the original RT-TDDFT. Importantly, we demonstrate with the QM9 dataset that the models trained on diverse molecules can generalize well to unseen molecules. In Figure 2 (a), we added an overview of the time propagation process, as well as an illustration about the one-step mapping detail using OrbEvo.
>
> **W2**. The units are missing from all the performance numbers in the paper.
>
> Thank you for raising this point. We are working on the localized atomic orbital basis, so the wavefunction coefficient itself is unitless, satisfying the normalization condition of $C^\\dagger\_n S C\_n \= 1$. So the $\\ell\_2$-MAE of coefficients is unitless. The other nRMSE metrics are relative error, i.e., the norm of the error divided by the norm of the target. So they also do not have units. We added one sentence to discuss the units at the beginning of Section 4.3.1 in the revised paper. We also added one sentence about the units for dipole and absorption spectra in Figure 3’s caption.
>
>
> **W3**. Neither the source code nor the dataset are provided. This renders the paper hard to follow.
>
> We will add links to the code and data in the paper once the paper is published, and we added a statement about this at the end of the introduction section in the revised paper. To ease the understanding about the implementation, we revised Figure 2 to include more illustrations about the model.
>
> **Q1**. About motivation: Could you provide a clearer picture of the framework? For example, how is the learned model used in RT-TDDFT to accelerate the computation?
>
> The usage of the learned model in RT-TDDFT framework is explained in the reply to **W1**. The speed-up mainly comes from two sources. First, the learned model leverages efficient neural network inference with GPU acceleration and does not need to solve the time-dependent TDDFT equation, which involves time-consuming operations such as numerical integrations. Second, the deep learning model learns to make predictions with a coarsened time intervals (10x downsampling ration used in our work), together with the time-bundling techniques, the neural network can produce the entire trajectory in 13 auto-regressive rollout steps, as opposed to the numerical simulation, which requires 1000 steps.

---

> ### Author Response · Authors · 2025-11-27
>
> **Q2**. About performance: How much acceleration is achieved? How should one interpret the metrics in Table 1 and Table 2?
>
> Thank you for the advice. In the updated paper, we include the computational cost in Appendix C which includes the training and inference times & GPU memory usages of OrbEvo models, as well as the simulation time using the classical solver. The computational cost is referred to in Section 4.3.1.
>
> Given only the molecular structure, the time we need for using OrbEvo to produce the entire trajectory is the “Ground-state DFT” time in Table 6 \+ the “Wavefunction \+ Property” time in Table 5\. While the time we need using classical numerical solvers is the “Total” time in Table 6\. We note that OrbEvo models can significantly accelerate the rollout during inference. For example, for OrbEvo-DM on QM9 dataset, the time for each molecule is 26.74 seconds \+ 73.1 seconds \= 99.8 seconds, while the time for classical solver is 3.2 hours.
>
> The metrics in Table 1 and 2: the $\\ell\_2$-MAE errors reflect the absolute error in wavefunction coefficients, and we also use 1-step $\\ell\_2$-MAE loss (after input normalization) for training. On the other hand, the nRMSE loss is the ratio between the error’s norm and the target’s norm. It is easier to interpret since it is normalized. For example, a nRMSE of 0.1 for dipole-$z$ means the dipole-$z$ has 10% error compared to the ground truth through the entire unrolled time steps (flatten the signal's time dimension to get a long vector and compute its norm).
>
> **Q3**. How does the model compare to direct property prediction methods?
>
> That is an interesting comparison. We conducted an additional experiment with direct property prediction. We adapt the OrbEvo-DM-sall model to take the molecular structure and the ground-state wavefunction coefficients as input and directly output the dipole along $z$ axis of all time steps, or the absorption spectra of all frequencies. Concretely, the dipole $z$ target is a 100-dimensional vector, and the absorption spectra is a 245-dimensional vector. We condition the model on the downsampled 100 step electric field to break the symmetry. To avoid potential conflict between these two targets, we train separate models for different properties (dipole and absorption), i.e., two models per dataset. We use the nRMSE loss as training loss, as well as the evaluation objective. We multiply the dipole $z$ target by 100 to normalize the target. We train the models for 12.5k and 25k iterations for MDA and QM9 respectively, and compare with the metrics of OrbEvo-DM-s8 model from the main paper. The results are summarized in the below table.
>
> | Model \\ Target | QM9 Dipole $z$ nRMSE | QM9 Absorption nRMSE | MDA Dipole $z$ nRMSE | MDA Absorption nRMSE |
> | :---- | :---- | :---- | :---- | :---- |
> | OrbEvo-DM-sall-property | 0.3640 | 0.1249 | 0.5196 | 0.1630 |
> | OrbEvo-DM-s8 | **0.1459** | **0.0752** | **0.2326** | **0.0671** |
>
> We observe that the direct property prediction does not perform as well as the wavefunction propagation model. We checked that the training losses are close to zero (QM9: dipole $z$ training loss \~0.019, absorption training loss \~0.007. MDA: dipole $z$ training loss \~0.011, absorption training loss \~0.005). Note that for the wavefunction propagation model, we do not have any direct supervision on the dipole and absorption targets. The cause for the property prediction model’s low accuracy may be due to the fact that although properties close to the ground states can be easy to learn, properties farther away in time may be harder to directly infer from the initial state as the space for possible dynamics can expand with time. Another reason is that the number of training pairs becomes significantly smaller if we only use the initial state and the final property as training pairs. These results suggest that by learning from the temporal evolution between consecutive time steps, we can effectively increase the amount of training data, thus improving the data efficiency.

---

> > ### Comment · Reviewer_ce9Z · 2025-11-28
> >
> > Thank you for the detailed response. The new revisions, clarifications and experimental results have addressed most of my concerns, and I would like to increase my rating as soon as possible (it seems that we are not able to edit our original review for now).

---

> > > ### Author Response · Authors · 2025-12-03
> > >
> > > We appreciate your quick reply to our rebuttal. Thank you again for reviewing our work!

---

### Official Review · Reviewer_6czi · 2025-10-31

**Soundness:** 3
**Presentation:** 3
**Contribution:** 3
**Rating:** 6
**Confidence:** 4

**Summary:**

This paper proposes *Orbital Transformers*, an equivariant graph Transformer designed to directly predict the *time evolution of Kohn–Sham wavefunctions* in real-time time-dependent density functional theory (RT-TDDFT). Unlike prior approaches that predict energies, Hamiltonians, or spectral observables, this model learns the mapping $C(t) \to C(t+\Delta t)$ (or $\Delta C_t$) directly, effectively learning the quantum propagation operator. The authors introduce an SO(2)-equivariant attention mechanism that takes the external electric field direction as the reference axis, and use FiLM-style conditioning to inject both the field’s direction and time-dependent amplitude. A local autoregressive temporal modeling scheme, along with pushforward training, enables the model to track the dynamic evolution of the system stably over several femtoseconds. Experiments on RT-TDDFT trajectories of QM9 and MD17 molecules under external fields show that the model accurately reproduces dipole dynamics and orbital evolution.

**Strengths:**

1. This is the first work to *directly predict the time evolution of wavefunctions* in RT-TDDFT rather than energies or Hamiltonians. The idea of learning an implicit propagation operator $f_\theta: C(t)\mapsto C(t+\Delta t)$ is conceptually novel and impactful.
2. Modeling system evolution under an external field is *physically meaningful* and directly corresponds to realistic nonequilibrium dynamics.
3. The paper presents an *innovative autoregressive temporal modeling* strategy that allows the neural network to continuously track and predict the system’s electronic evolution, combining local ΔC prediction with pushforward training to reduce error accumulation.
4. The *integration of field information into the atomic graph network* is well designed: the electric-field direction defines the SO(2) equivariant reference axis, and the field information enters through FiLM-style modulation. This construction is both physically grounded and computationally efficient.

**Weaknesses:**

1. The network predicts the next-step evolution solely from the current state, **and although the authors adopt certain stabilization strategies** — such as local autoregressive temporal modeling, pushforward training, and ΔC prediction — these mechanisms only address short-term error accumulation. **There remains no explicit architectural component (e.g., temporal attention or recurrent memory) to model long-range temporal dependencies**, which consequently limits robustness and stability during long-horizon propagation beyond several femtoseconds.

2. The SO(2)-equivariant design relies on the external-field direction as the rotational reference axis. Once the external field vanishes, this reference loses physical meaning, and the network no longer has a well-defined axis for the equivariant operations. Moreover, if the external-field direction varies over time or differs across samples, the reference frame for the SO(2) operations also changes with time. It is unclear how well the model would perform under such cases.

3. The paper appears to **lack systematic efficiency evaluation experiments**, for example, comparisons on runtime and computational resource usage, which would be important to assess the practical utility of the proposed model.

**Questions:**

The authors generate and RT-TDDFT trajectories with external fields for standard datasets like QM9 and MD17, which may enrich the data resources for time-dependent electronic-structure ML. Will the generated trajectories and other data be publicly released, providing valuable data for future time-dependent quantum ML research?

---

> ### Author Response · Authors · 2025-11-27
>
> Thank you for taking the time to read and evaluate our paper,  and thank you for the patience in waiting for our response. We reply to your comments and questions in the following.
>
> **W1**. The network predicts the next-step evolution solely from the current state, and although the authors adopt certain stabilization strategies — such as local autoregressive temporal modeling, pushforward training, and ΔC prediction — these mechanisms only address short-term error accumulation. There remains no explicit architectural component (e.g., temporal attention or recurrent memory) to model long-range temporal dependencies, which consequently limits robustness and stability during long-horizon propagation beyond several femtoseconds.
>
> In this paper, we mostly follow the one-step training, which currently is the most commonly adopted time propagation strategy in the machine learning time-dependent PDE literature (such as fluid dynamics simulation). The motivation of this strategy may root from the Markov assumption in learning the dynamics propagation \[1\]. E.g., the mappings between consecutive time steps are similar and transferable across different starting times, and only depend on the immediate historical steps. We observe a similar pattern in our orbital based TDDFT problem, where the onestep error stays practically constant across different starting times. Additionally, we use the time bundling technique, which groups multiple time steps as one step and makes the model to condition on several historical steps (8 in our implementation).
>
> On the other hand, we are aware of papers that make use of temporal attention \[2\] and recurrent training \[3, 4\]. However, such techniques may require significantly increased GPU memory usage. More generally, achieving long-term rollout is indeed an important problem and is an active research topic. E.g., there also exist techniques that make use of diffusion models to achieve long rollout \[5\], which we tested during our development but failed to make it work for our problem, potentially due to the high dimensionality of the wavefunction coefficients. In our work, our main goal is to identify a minimum set of techniques for the machine learning model to produce wavefunction predictions that are accurate enough to produce correlated properties without directly supervising the properties. In future work, more advanced techniques may be explored to produce more stable rollout for a longer period of time.
>
> \[1\] Tran, Alasdair, et al. "Factorized Fourier Neural Operators." The Eleventh International Conference on Learning Representations.
> \[2\] Han, Xu, et al. "Predicting Physics in Mesh-reduced Space with Temporal Attention." International Conference on Machine Learning (ICML 2022). Vol. 39\. 2022\.
> \[3\] List, Bjoern, et al. "Differentiability in unrolled training of neural physics simulators on transient dynamics." Computer Methods in Applied Mechanics and Engineering 433 (2025): 117441\.
> \[4\] Kochkov, Dmitrii, et al. "Machine learning–accelerated computational fluid dynamics." Proceedings of the National Academy of Sciences 118.21 (2021): e2101784118.
> \[5\] Lippe, Phillip, et al. "PDE-Refiner: Achieving Accurate Long Rollouts with Neural PDE Solvers." Advances in Neural Information Processing Systems 36 (2023): 67398-67433.

---

> ### Author Response · Authors · 2025-11-27
>
> **W2**. The SO(2)-equivariant design relies on the external-field direction as the rotational reference axis. Once the external field vanishes, this reference loses physical meaning, and the network no longer has a well-defined axis for the equivariant operations. Moreover, if the external-field direction varies over time or differs across samples, the reference frame for the SO(2) operations also changes with time. It is unclear how well the model would perform under such cases.
>
> This is a very interesting point. Indeed, the electric field in our case vanishes to zero at later time steps. However, we think it is still reasonable to model the entire trajectory as an SO(2)-equivariant process. In fact, at ground state without any external electric field, the system is SO(3)-equivariant. When an external time-dependent electric field is applied, the system will become SO(2)-equivariant due to the induced oscillation in the excited states. Even after the field is removed after some period of time, the system is still maintaining the excited state dynamics. In this case, the system will not go back to SO(3)-equivariance unless a relaxation process is introduced. In the current RT-TDDFT, there is no relaxation, hence the process will not regain SO(3) equivariance.
>
> On the other hand, it is also possible extend our SO(2)-layernorm to have the desired behavior, i.e., when the external electric field goes to zero, the model can regain the SO(3)-equivariance, which may be in a better agreement with the one-step physical process. We can achieve this by altering the computation of the scaling and bias in the SO(2)-layernorm. In particular, using notations from Section 3.2.5 in the revised paper, we can define the new conditioning as $y\_\\ell=(1 \+ s\_\\ell) \\odot LN(x)\_\\ell \+ b\_\\ell$, where $s\_\\ell$ and $b\_\\ell$ are computed from the electric field intensities at current and next time steps as $\[s\_\\ell, b\_\\ell\] \= MLP(E(t)) \\odot (W E(t))$, where $W E(t)$ is a linear transformation of $E(t)$ without adding bias. In this way, when the electric field $E(t)$ becomes zero, $s\_\\ell$ and $b\_\\ell$ will also be zero, and the new conditioning reduces to the original layernorm in EquiformerV2 and is SO(3)-equivariant. We implemented this method based on OrbEvo-DM-s8 and trained a model on the MDA dataset, the new model is dubbed OrbEvo-DM-s8-zerocond. We compare it with the OrbEvo-DM-s8 model from the main paper. The test results are summarized in the below table.
>
> | Model | 1-step $\\ell\_2$-MAE | Rollout nRMSE | Dipole $z$ nRMSE | Absorption nRMSE |
> | :---- | :---- | :---- | :---- | :---- |
> | OrbEvo-DM-s8-zerocond | 0.0244 | 0.1830 | 0.2437 | 0.0734 |
> | OrbEvo-DM-s8 | **0.0242** | **0.1778** | **0.2326** | **0.0671** |
>
> We observe that the new conditioning method can give close but not better results. This can be explained by the aforementioned reasoning regarding the SO(2)-symmetry of the trajectory and the reasoning that although the one-step physics may be SO(3)-equivariant after the electric field vanishes, the SO(2)-equivariance alone may be sufficient to describe the symmetry in the data distribution. Overall, we believe the new conditioning design may be valuable in more sophisticated scenarios, such as when the electric field can change direction in time. Nevertheless, for our case where the electric field is always along one pre-defined direction, we believe that our current design in the main paper can already serve the purpose well, which is also demonstrated by the WF-sall-inv-cond ablation study (last row of Table 3 in the revised paper),

---

> ### Author Response · Authors · 2025-11-27
>
> **W3**. The paper appears to lack systematic efficiency evaluation experiments, for example, comparisons on runtime and computational resource usage, which would be important to assess the practical utility of the proposed model.
>
> Thank you for the advice. In the updated paper, we include the computational cost in Appendix C which includes the training and inference times & GPU memory usages of OrbEvo models, as well as the simulation time using the classical solver. The computational cost is referred to in Section 4.3.1.
>
> Given only the molecular structure, the time we need for using OrbEvo to produce the entire trajectory is the “Ground-state DFT” time in Table 6 \+ the “Wavefunction \+ Property” time in Table 5\. While the time we need using classical numerical solvers is the “Total” time in Table 6\. We note that OrbEvo models can significantly accelerate the rollout during inference. For example, for OrbEvo-DM on QM9 dataset, the time for each molecule is 26.74 seconds \+ 73.1 seconds \= 99.8 seconds, while the time for classical solver is 3.2 hours.
>
> **Q1**. The authors generate and RT-TDDFT trajectories with external fields for standard datasets like QM9 and MD17, which may enrich the data resources for time-dependent electronic-structure ML. Will the generated trajectories and other data be publicly released, providing valuable data for future time-dependent quantum ML research?
>
> Yes, we will release code and data upon publication. We added a statement about this at the end of the introduction section in the revised paper.
>
> **S4**. The integration of field information into the atomic graph network is well designed: the electric-field direction defines the SO(2) equivariant reference axis, and the field information enters through FiLM-style modulation. This construction is both physically grounded and computationally efficient.
>
> Thank you for the kind words. In Figure 2 of the revised paper, we added a figure of the SO(2)-LayerNorm to better illustrate this process. We also tested the SO(2)-equivariance in Appendix F.

---

### Official Review · Reviewer_F1P1 · 2025-11-01

**Soundness:** 3
**Presentation:** 3
**Contribution:** 3
**Rating:** 4
**Confidence:** 5

**Summary:**

The paper proposed a new model and method that learns the time-dependent DFT's properties, and has shown that the new proposed method, combined with a serious method improvement, can predict nicely the properties from TDDFT.

**Strengths:**

1. The network is well designed with very good respect for the physics.
2. The increase in accuracy is very impressive.

**Weaknesses:**

1. The writing needs more clarification, for example, the model design, more words in the caption and the method part can be helpful to understand it clearly.
2. The experimental work is comparably weak. More comparison with other models, and specific studies on in-distribution, out-of-distribution systems can make this study more solid.

**Questions:**

1. In real applications, we often focus on a specific kind of system, where we often do not have that much data. In terms of this, how is this method transferable to similar structures but with limited training data? How is the data efficiency in that?

2. How to transfer the model to different external field conditions?

3. How is the SO3-SO2 mapping achieved in layer norm? I haven't found it in the paper.

---

> ### Author Response · Authors · 2025-11-27
>
> Thank you for taking the time to read and evaluate our paper, and thank you for the patience in waiting for our response. We reply to your concerns and questions in the following.
>
> **W1**. The writing needs more clarification, for example, the model design, more words in the caption and the method part can be helpful to understand it clearly.
>
> Thank you for pointing this out. We uploaded a revised version of the paper where we improved writing without changing the method or results. In the revised PDF, we updated **Figure 2** to include an illustration of the overall model pipeline, the model architecture, density matrix featurization via tensor contraction, as well as details about electric field conditioning. We also expanded its caption to have more explanation. We also updated the writing of the method part, including the description of the electric field conditioning and the electronic state sampling for the OrbEvo-DM model. We also updated Figure 3 to include one more test example and improved the presentation. We apologize for not including the above information in the initial submission. Please let us know if anything is not clear so that we could explain it here and update the paper if needed.
>
> **W2**. The experimental work is comparably weak. More comparison with other models, and specific studies on in-distribution, out-of-distribution systems can make this study more solid.
>
> We would like to respectfully push back on the reviewer's impression of the experimental work being weak. We generated our own datasets using hundreds of thousands of CPU hours. We tested two variants of OrbEvo on both data with the same molecule but with different geometries (MDA), and more importantly data with diverse molecules  (QM9), which validates the generalization ability of the proposed approach. Moreover, we extensively validated our design choices through ablation studies on the MDA dataset where the results were in the appendix. To improve the clarity, we moved the ablation studies from appendix to the main text, which would enrich the experimental section in the main text. We apologize if the previous organization may cause confusion about the overall experimental study.
>
> Regarding models, as machine learning time-dependent PDEs (such as fluid dynamics or weather forecasting) gains broader attention rather recently, learning TDDFT is a relatively new task, especially using atomic-orbital-based modeling. As a result, we are not aware of existing models that can be directly applied to such a problem. To showcase the ability of learning such a problem we designed our own models based on EquiformerV2, which is a state-of-the-art model in learning ground-state DFT properties and makes use of equivariant features and interactions, which can be important since we are predicting wavefunction with rich equivariant structures. Nevertheless, we agree that we did not fully explore the model space, such as architecture choices, which is a limitation of our current work. However, doing so would require substantial effort and may not outperform the current EquiformerV2-based model. As shown in the EquiformerV2 paper \[1\], it is able to achieve comparable or better results against other equivariant baseline models. Thus we leave the model exploration in future work.
>
>
> We agree with the reviewer that specific studies on in-distribution (ID), out-of-distribution (OOD) systems can improve the experimental study. Using our generated 5000 QM9 data, we created an OOD split for it based on the number of atoms in a molecule. In particular, we use the number of atoms from 8 to 20 for training (3955 samples), number of atoms \= 21 for validation (518), and number of atoms from 23 to 29 for testing (527 samples). We train the OrbEvo-DM-s8 model using this OOD split, the trained model is dubbed OrbEvo-DM-s8-ood. In the below table, we evaluate the trained model on the validation and test sets of the OOD split.
>
> | Model | Dataset | 1-step $\\ell\_2$-MAE | Rollout nRMSE | Dipole $z$ nRMSE | Absorption nRMSE |
> | :---- | :---- | :---- | :---- | :---- | :---- |
> | OrbEvo-DM-s8-ood | QM9\_ood \- val | 0.0142 | 0.1498 | 0.1113 | 0.0615 |
> | OrbEvo-DM-s8-ood | QM9\_ood \- test | 0.0132 | 0.1482 | 0.1175 | 0.0697 |

---

> > ### Author Response · Authors · 2025-11-27
> >
> > Rather surprisingly, the OOD split gives better average test accuracy than the random split reported in our main paper (Rollout nRMSE \= 0.1885, Dipole z nRMSE \= 0.1459, Absorption nRMSE \= 0.0752). However, the numbers are not directly comparable since different test data are used. To fairly compare the ID and OOD performance, we evaluate the models on the validation and test data that are both in the OOD split and the random split. Specifically, there are 67 molecules in the intersection of the OOD validation set and the random validation set (QM9\_id\_ood\_intersect \- val), and 43 molecules in the intersection of the OOD test set and random test set (QM9\_id\_ood\_intersect \- test). We report the results of both the model trained on the OOD split (OrbEvo-DM-s8-ood) and the model trained using the random split (OrbEvo-DM-s8 in the main paper) in the below table.
> >
> > | Model | Dataset | 1-step $\\ell\_2$-MAE | Rollout nRMSE | Dipole $z$ nRMSE | Absorption nRMSE |
> > | :---- | :---- | :---- | :---- | :---- | :---- |
> > | OrbEvo-DM-s8-ood | QM9\_id\_ood\_intersect \- val | **0.0141** | **0.1500** | **0.1126** | 0.0633 |
> > | OrbEvo-DM-s8 | QM9\_id\_ood\_intersect \- val | 0.0146 | 0.1579 | 0.1153 | **0.0628** |
> > | OrbEvo-DM-s8-ood | QM9\_id\_ood\_intersect \- test | **0.0137** | **0.1496** | 0.1142 | 0.0666 |
> > | OrbEvo-DM-s8 | QM9\_id\_ood\_intersect \- test | 0.0139 | 0.1521 | **0.1074** | **0.0636** |
> >
> > Despite the randomness due to the small amount of common val / test data, we can see that overall the model trained on the random split performs closer to the model trained on the OOD split. Overall, the above results show that the OrbEvo model is able to generalize on larger systems than those in the training data. The results also suggest that larger systems may not necessarily be more challenging for learning the dynamics of wavefunctions, potentially due to the fact that smaller systems can exhibit dynamic patterns with a larger magnitude or more complex behaviors. We will include these results in the revised paper. Thank you for suggesting this study.
> >
> > \[1\] Liao, Yi-Lun, et al. "EquiformerV2: Improved Equivariant Transformer for Scaling to Higher-Degree Representations." The Twelfth International Conference on Learning Representations.

---

> > > ### Author Response · Authors · 2025-11-27
> > >
> > > **Q1**. In real applications, we often focus on a specific kind of system, where we often do not have that much data. In terms of this, how is this method transferable to similar structures but with limited training data? How is the data efficiency in that?
> > >
> > > Thank you for raising this interesting point. One typical way to handle the small-data regime is through pretraining and finetuning, where we use a model pretrained on a large scale dataset and fine-tune the model on the small dataset. To illustrate this process, we use the OrbEvo-DM-s8 pretrained on the QM9 dataset and fine-tune it on a MDA subset with 200 molecules. We validate on a 100 molecule subset and test on the original 500 test set. We test three scenarios: (1) directly using the QM9 model to predict on the test set (QM9-zero-shot), (2) training from scratch for 100k iterations, and (3) finetuning from the model pretrained on QM9 for 100k iterations. We report the results in the below table.
> > >
> > > | Model | 1-step $\\ell\_2$-MAE | Rollout nRMSE | Dipole $z$ nRMSE | Absorption nRMSE |
> > > | :---- | :---- | :---- | :---- | :---- |
> > > | QM9-zero-shot | 0.0746 | 0.7396 | 0.7035 | 0.3054 |
> > > | Training from scratch | 0.0442 | 0.3631 | 0.4514 | 0.1402 |
> > > | Finetuning from QM9 | **0.0274** | **0.2312** | **0.2064** | **0.0757** |
> > >
> > > We can see that although QM9-zero-shot does not perform well, possibly due to the distribution shift from QM9 to MDA (e.g., number of atoms), finetuning from QM9 results in better accuracy than training from scratch, showing that using a pretrained model helps the convergence on finetuning, which can consequently help the data efficiency.
> > >
> > > **Q2**. How to transfer the model to different external field conditions?
> > >
> > > In general, a time-dependent electric field has 3 cartesian components ($E\_x(x, y, z, t)$, $E\_y(x, y, z, t)$, $E\_z(x, y, z, t)$) which can be complex and carry phase. In our framework, our model is conditioned on the current and next time bundles (several time steps), to handle different external electric fields, we can flatten different components and concatenate them into channel dimensions then apply temporal convolution and pooling to produce an encoding vector. We can then use this vector to condition the propagation model. However, existing deep learning models most commonly condition on scalar valued information (such as in diffusion models), there may be unique challenges to extend to function valued conditioning and we believe it is an important and interesting future direction to extend our current method.
> > >
> > > **Q3**. How is the SO3-SO2 mapping achieved in layer norm? I haven't found it in the paper.
> > >
> > > It is achieved by adding a bias to the layernorm outputs. The bias is set such that it only has nonzero elements for $m=0$ and 0 otherwise. For example, for the $\\ell=2$ feature, it’s a 5 dimensional vector corresponding to $m=-2, \-1, 0, 1, 2$, we will only add bias to the $m=0$ component. The reason is that the electric field is along the $z$ direction and the spherical harmonics projection of it will only have a non zero component for $m=0$. In this way if the molecule is rotated around the $x$ or $y$ axis but the electric field direction is not changed, the equivariance will no longer hold since the added bias will not change.
> > >
> > > We extended the description of 3.2.5 in the paper. We also added a figure to illustrate the SO(2)-LayerNorm operation in Figure 2(f). We apologize for the lack of description in the initial version. We also tested the SO(2)-equivariance in Appendix F.

---

### Official Review · Reviewer_GR5K · 2025-11-01

**Soundness:** 3
**Presentation:** 4
**Contribution:** 3
**Rating:** 8
**Confidence:** 4

**Summary:**

This paper introduces OrbEvo, an equivariant graph transformer framework for learning the time evolution of Kohn–Sham wavefunctions in real-time time-dependent density functional theory (RT-TDDFT). Unlike prior works such as OrbFormer, which focus on static ground-state properties, OrbEvo aims to learn the dynamics of electronic states under external electric fields.
The authors propose two model variants: OrbEvo-FullWF, which aggregates wavefunction features through pooling across occupied states, and OrbEvo-DM, which computes density-matrix-based interactions between states via tensor contraction. The model employs SO(2)-equivariant conditioning to represent field-induced symmetry breaking and a pushforward training scheme to stabilize long-horizon rollout. Experiments on QM9 and MD17 demonstrate that OrbEvo-DM outperforms the pooling-based variant, capturing physically consistent time-dependent dipole moments and absorption spectra.

**Strengths:**

**Easy to follow and clear motivation**: The paper provides an intuitive and well-structured explanation of the challenge in modeling time-dependent DFT, with smooth transitions from motivation to formulation.

**Tackling an impactful and general problem**: The work addresses a scientifically meaningful and practically impactful challenge: learning the time evolution of Kohn–Sham wavefunctions to accelerate quantum dynamics simulations. Importantly, the model is trained across thousands of diverse molecules (QM9) and demonstrates generalization to unseen molecular systems, showing potential as a shared, cross-molecular surrogate model for electronic dynamics. This highlights the scalability and versatility of the approach beyond single-molecule modeling.

**Sound model design**: The use of SO(2)-equivariant conditioning, density-matrix features, and push forward training demonstrates strong physical insight and solid engineering.

**Weaknesses:**

Dataset scale is limited, making it difficult to assess generalization to larger molecules or different field conditions.

**Questions:**

1. **Applicability to static DFT**: Can the OrbEvo architecture be used for static ground-state DFT tasks, such as predicting stationary wavefunction coefficients or density matrices? Or is it strictly limited to time-dependent TDDFT propagation?

2. **Ablation on time bundling**: How much does the time bundling technique contribute to the model’s performance and stability? It would be helpful to see results comparing models with and without time bundling.

---

> ### Author Response · Authors · 2025-11-27
>
> Thank you for taking the time to read and evaluate our paper. We appreciate your positive impressions on paper structure, task’s impact, and model design and engineering. We reply to your concerns and questions in the following.
>
> **W1**. Dataset scale is limited, making it difficult to assess generalization to larger molecules or different field conditions.
>
> We agree with the reviewer about the dataset scale and different (electric) field conditioning, and we aim to explore them in future work.
>
> We agree that the systems we tested (MDA, QM9) are not large, and assessing the generalization to larger molecules remains challenging. However, given the success in large-scale foundation models for molecular systems, which largely uses design choices developed on smaller systems. Larger systems also provide richer information inside each system, providing more learning data for the model to capture orbital interactions. We believe that the techniques developed in this work are also applicable to larger systems.
>
> On the other hand, developing methods that can work with various field conditions is important to let ML models learn the full capacity of TDDFT simulations. While the electric pulse considered in this paper is simple, it allows us to study the excited electronic states within the linear response regime for various molecules with different structures and compositions, which is one of the major use cases of TDDFT simulation. Another important use case of TDDFT simulation is to use a stronger electric field which can have a more complicated temporal profile. For such an application, conditioning on different electric fields is indeed critical. However, developing a conditioning method for diverse electric fields will increase the dimensionality of the input space and may face new sets of challenges such as generalization to unseen conditions.
>
> In this work we aim to showcase that based on techniques from ML for time-dependent PDE and ML for molecules as well as specially designed approaches, a deep learning model is able to achieve an accuracy level that is good enough to derive useful downstream properties without any direct supervision on those properties. Thus, we position it as an important direction to be explored and leave it for future work.
>
> **Q1**. **Applicability to static DFT**: Can the OrbEvo architecture be used for static ground-state DFT tasks, such as predicting stationary wavefunction coefficients or density matrices? Or is it strictly limited to time-dependent TDDFT propagation?
>
> Thank you for raising this very interesting question. Yes, the OrbEvo architecture can be used for static ground-state DFT. In the DFT SCF loop, an initial guess for the density matrix is performed before the loop, from which we can derive the wavefunction. We can use this initial guess for wavefunction as input to the model and predict the converged wavefunction, from which we can compute the ground-state density matrix. As shown in a recent paper \[1\], learning the mapping from initial guess to the converged state helps the performance.
>
> \[1\] Yin, Shi, et al. "Advancing Universal Deep Learning for Electronic-Structure Hamiltonian Prediction of Materials." arXiv preprint arXiv:2509.19877 (2025)

---

> > ### Author Response · Authors · 2025-11-27
> >
> > **Q2**. **Ablation on time bundling**: How much does the time bundling technique contribute to the model’s performance and stability? It would be helpful to see results comparing models with and without time bundling.
> >
> > This is an interesting ablation study. We additionally conduct an experiment on the MDA dataset with various time bundle sizes 1, 2, 4, 16 and with 8 (used in the main paper) where a time bundle size 1 corresponds to without time bundling. We report 1-step error (average error for the time bundle), rollout errors of different trajectory lengths (start from time 0, produce trajectories with lengths 8, 16, 32, 64, 100, larger bundle needs less autoregressive steps), as well as relative errors for dipole-$z$ and absorption on the full rollout. The test results are summarized in the below table:
> >
> > | Time bundle size | 1-step $\\ell\_2$-MAE | 8-step Rollout nRMSE | 16-step Rollout nRMSE | 32-step Rollout nRMSE | 64-step Rollout nRMSE | 100-step Rollout nRMSE | Dipole $z$ nRMSE | Absorption nRMSE |
> > | :---- | :---- | :---- | :---- | :---- | :---- | :---- | :---- | :---- |
> > | 1 | **0.0093** | 0.0780 | 0.0340 | 0.1363 | 0.4433 | 0.9032 | 0.9526 | 0.1684 |
> > | 2 | 0.0130 | **0.0668** | 0.0340 | 0.1106 | 0.3087 | 0.5765 | 0.5669 | 0.1228 |
> > | 4 | 0.0139 | 0.0693 | **0.0289** | **0.0572** | **0.1235** | 0.1979 | 0.2670 | 0.0758 |
> > | 8 | 0.0242 | 0.0720 | 0.0378 | 0.0677 | 0.1245 | **0.1778** | **0.2328** | **0.0672** |
> > | 16 | 0.0588 | 0.1026 | 0.0544 | 0.1075 | 0.2040 | 0.2872 | 0.2641 | 0.0922 |
> >
> > We observe that using a smaller time bundle size results in a smaller 1-step error. This is because the model needs to predict fewer steps, and closer steps in time are easier to predict. For rollout errors, time bundle sizes of 1 and 2 can produce correlated rollouts at 16 steps, but start to diverge for 32 or more steps. Time bundle size of 4 performs well for 32 steps, but becomes less effective than time bundle size 8 for longer rollout. Time bundle size 8 produces the best wavefunction in the overall rollout. Time bundle size of 16 stays stable but the accuracy is not as good as 8 or 4\.
> >
> > In terms of training, we observe that time bundle size 1 and 2 start to overfit to onestep mapping and the validation rollout errors start to overfit during the training, with the best validation 100-step rollout error of around 0.4 occurs at around 100k iteration for time bundle size 1, and around 0.6 at around 120k iterations for time bundle size 2 (total training iteration is 300k). Moreover, we observe some oscillations in the validation rollout curves for time bundle size 4 during training.
> >
> > Note that during training we randomly sample from all 100 steps for feasible starting time for onestep training. So the total number of training pairs are similar for different time bundle sizes. Overall, a time bundle size of 8 remains a reasonable choice, in which case the model needs to unroll 13 steps to produce the entire 100-step trajectory. We will include this study in the revised paper. Thank you for the suggestion.

---

### Meta-Review · Area_Chair_9cwh · 2026-01-05

**Summary:**

The authors present a neural network which they claim can be used to accelerate the forward time evolution of real-time TDDFT, a method for simulating the time evolution of the electronic structure of molecules excited above the ground state. They generate data from molecules in the QM9 dataset and compare their method against standard TDDFT calculations with favorable results. The authors agreed the work was of interest and the results were good, albeit with some concerns about the scale of the data being relatively small. Most reviewers recommend acceptance and I agree.

I would have liked to see more discussion of the limitations of TDDFT. While it is a commonly used method, it is also known to be an approximation with many accuracy issues - for instance, TDDFT cannot represent conical intersections, where the ordering of excited state energies switches. Improving the accuracy of density functional methods for time dependent systems would be a high-impact future direction to pursue.

**Reviewer Concerns:**

Some reviewers were concerned about whether the comparisons were general enough, given the small number of systems investigated. There were also many implementation questions related to SO(2) layer norm and other details. The authors generally did a good job of addressing these concerns.

**Reviewer Scores:**

I think the one reviewer who gave a reject rating might have revised their rating up slightly. Reviewer ce9Z explicitly said they would increase the score if they could but the score is frozen.

---

### Decision · Program_Chairs · 2026-01-26

Accept (Poster)